# Synergistic effects of notoginsenoside R1 and saikosaponin B2 in atherosclerosis: A novel approach targeting PI3K/AKT/mTOR pathway and macrophage autophagy

Yihua Wang[1,2], Qing Liao[1], Xue Mei[1], Han Xu[1,3], Lijun Luo[1], Rong Huang[1], Yue Tang[1], Chunyang Zhou[iD][1]*

**1** Institute of Materia Medica, School of Pharmacy, North Sichuan Medical College, Nanchong, Sichuan, China, **2** Department of Pharmacy, Nanbu People's Hospital, Nanbu, Sichuan, China, **3** Southwest Medical University Zigong Affiliated Hospital, Zigong Mental Health Center, Zigong, Sichuan, China

* zhouchunyang@nsmc.edu.cn

## Abstract

Atherosclerosis (AS) is a major global contributor to cardiovascular diseases and associated mortality. The traditional treatment primarily relies on statins; however, these drugs are often linked to side effects such as liver dysfunction and muscle impairment. Recent studies have highlighted the potential protective properties of saponin compounds derived from traditional herbal sources, such as notoginsenoside R1 (NGR1) and saikosaponin B2 (SSB2), in combating atherosclerosis. However, the comprehensive effects of these compounds against atherosclerosis and their underlying mechanisms remain inadequately understood. Firstly, we employed network pharmacology analysis to identify 113 common targets, including mTOR and CASP3, for NGR1, SSB2, and atherosclerosis from databases such as TCMSP. We constructed a protein-protein interaction (PPI) network and performed GO and KEGG enrichment analyses, whcih revealed key signaling pathways involved in PI3K/Akt, inflammation, and autophagy. The atherosclerosis model was established using ApoE$^{-/-}$ mice fed with a "Western diet" followed by treatment with NGR1, SSB2, or NS combination. Histological examinations, including hematoxylin-eosin (HE) staining, oil red O (ORO) staining, and CD68 immunofluorescence, were conducted to evaluate the pathological conditions of the aortic root, liver and kidneys in ApoE$^{-/-}$ mice. Our findings demonstrate that the NS combination improves lipid levels, lipid transport, and unstable plaque formation in ApoE$^{-/-}$ mice without adverse effects on liver or kidney function. Finally, oxidized low-density lipoprotein (ox-LDL) was used to culture RAW264.7 macrophages to establish an *in vitro* foam cell model. The effects of NS combination on lipid uptake, inflammatory response, apoptosis, the PI3K/Akt/mTOR signaling pathway, and autophagy were evaluated using methods such as CCK-8 assay, Oil Red O staining, immunofluorescence analysis, flow cytometry, RT-PCR,

**Data availability statement:** All figures files are available from the dryad database (http://datadryad.org/share/4GvB7LlOnZ0wtL8Og2WI-plaC9ZctRnAeJkkWt7K1Fnl).

**Funding:** 1.The Scientific Startup Foundation for Doctors of North Sichuan Medical College(CBY23-QDA11) to XM; 2.Key Research and Development Projects of the Sichuan Provincial Department of Science and Technology (2020YFS0528) to CZ. The funders had no role in study design, data collection and analysis, decision to publish and preparation of the manuscript.

**Competing interests:** The authors have declared that no competing interests exist.

and Western blot analysis. The results indicated that NS combination promoted autophagy by inhibiting the PI3K/Akt/mTOR pathway. This significantly alleviated inflammation, reduced apoptosis, and decreasing lipid accumulation, thereby improving the pathological progression of atherosclerosis. Collectively, this study demonstrates, for the first time, that the NS combination synergistically activates macrophage autophagy by suppressing the PI3K/AKT/mTOR pathway, thereby attenuating lipid accumulation, inflammation, and apoptosis in atherosclerotic models.

## 1 Introduction

Atherosclerosis(AS) is one of the leading causes of cardiovascular diseases and related mortality globally, including conditions such as myocardial infarction and stroke [1,2]. With the aging global population and the increasing prevalence of unhealthy lifestyles, the incidence of atherosclerosis has been steadily increasing. Globally, millions of people die each year from atherosclerosis-related diseases, particularly in developing countries, where the mortality rate continues to climb [3]. Current therapeutic strategies for atherosclerosis are primarily centered around pharmacological interventions, including lipid-lowering agents, antiplatelet therapies, and invasive procedures like angioplasty or bypass surgery. However, these treatments often come with significant challenges in clinical practice, including issues such as drug resistance, hepatotoxicity, and an increased risk of severe hemorrhagic complications [4–6]. Given these limitations, there is an urgent need for the continuous development of novel therapeutic strategies and drug candidates to minimize adverse effects while enhancing clinical outcomes.

Recent studies have shown that phytochemicals derived from traditional herbal plants exhibit a wide range of biological activities, contributing to significant preventive and therapeutic effects against various diseases [7–9]. Among them, saponins from traditional Chinese herbal medicine are particularly notable due to their multifaceted roles in preventing and treating atherosclerotic diseases through mechanisms such as anti-inflammatory, antioxidant, cholesterol-lowering, immunomodulatory, and antiplatelet aggregation effects [10]. Notably, *Panax notoginseng* and *Bupleurum*, have demonstrated prominent efficacy in cardiovascular disease management.

Notoginsenoside R1 (NGR1) is a unique component of *Panax notoginseng*. Previous studies have shown that NGR1 exhibits a variety of biological activities and provides protective effects across multiple organs, including neuroprotection, cardiovascular protection, pulmonary protection, and hepatic protection [11]. Jia and his team [12]were the first to demonstrate that NR1 may counteract atherosclerosis by modulating inflammation, oxidative stress, and lipid metabolism. A recent study has further highlighted its role in ameliorating atherosclerosis by alleviating inflammatory responses and inhibiting endothelial dysfunction [13].

Saikosaponin is the active component in *Bupleurum*, account for up to 7% of the plant's dry root weight and possess a wide array of pharmacological activities. Among them, saikosaponin A (SSA) and saikosaponin D (SSD) have received the most

attention in pharmacological research. Of particular interest is saikosaponin B2 (SSB2), a compound formed when SSD is exposed to high temperatures or low pH, reflecting its significance in the traditional preparation of TCM decoctions [14,15]. Previous studies have indicated that SSB2 exerts anti-inflammatory effects in LPS-induced macrophages, and influence the lipid metabolism process in HepG2 cells, demonstrating anti-atherosclerotic activity potential [16,17]. These findings suggest SSB2 may regulate the occurrence and development of atherosclerosis, potentially serving as a valuable new drug for treating atherosclerosis. Although studies have suggested that NGR1 and SSB2 may have protective effects in cardiovascular diseases, the precise molecular mechanisms by which these compounds exert their effects on atherosclerosis, either individually or in combination, remain incompletely understood.

Current research attributes the development of atherosclerosis to lipid accumulation, chronic inflammation, and immune dysregulation [18]. However, the pathogenesis of this disease is not fully understood. A growing body of evidence links atherosclerosis progression to foam cell formation, which derived from monocyte differentiation and matured macrophage [19,20].When exposed to excessive ox-LDL, macrophages acquired a foam cell-like phenotype through scavenger receptors, leading to necrotic cores formation within atherosclerotic plaques [21]. Persistent exposure to atherogenic lipoproteins, the necrosis and apoptosis of these foam cells result in lipid core necrosis, activating vascular inflammatory responses that lead to cell apoptosis and abnormal lipid metabolism. In contrast, macrophage autophagy has been shown to inhibit the progression of atherosclerosis and stabilize plaques by promoting intracellular lipid degradation [22–24]. Enhancing autophagic activity in macrophages facilitates intracellular lipid degradation; while inhibition of autophagy predisposes cells to necrosis and inflammation following foam cell death, contributing to plaque formation and rupture [25]. Considering these insights, inhibiting the formation of macrophage-derived foam cells and enhancing macrophage autophagy represent effective therapeutic strategies for the treatment of atherosclerosis.

Despite some existing studies revealing the biological activity and potential therapeutic effects of NGR1 and SSB2 in atherosclerosis, there is currently no definitive conclusion regarding their efficacy when applied individually. More importantly, whether NGR1 and SSB2 can synergistically enhance therapeutic effects remains underexplored. Therefore, the combined use of these two substances with unique pharmacological properties not only inherits traditional Chinese medicine compatibility theories but also represents an innovative exploration in modern medical translation. This study aims to delve into the potential of combining NGR1 and SSB2 for atherosclerosis treatment based on contemporary understanding of the mechanisms underlying atherosclerosis development. Specifically, through a comprehensive approach incorporating network pharmacology, *in vitro* cellular experiments, and *in vivo* animal studies, we will systematically evaluate their mechanisms of action in atherosclerosis. The focus will be on how both agents may exert a synergistic effect by regulating the formation of macrophage-derived foam cells and influencing macrophage autophagy processes to improve therapeutic outcomes for atherosclerosis. We anticipate that this combination therapy strategy will not only effectively alleviate the pathological progression of atherosclerosis but also reduce common side effects associated with existing treatments, ultimately providing a safer and more effective new treatment option for clinical practice. Consequently, this research not only theoretically expands new avenues for atherosclerosis treatment but is also expected to offer more scientific evidence and practical guidance for the application of traditional Chinese medicine formulations.

## 2 Materials and methods

### 2.1 Drugs and reagents

Notoginsenoside R1 (NGR1, chemical structure $C_{47}H_{80}O_{18}$, molecular weight = 933 g/mol, purity = 99.5%, BD2143) and saikosaponin B2 (SSB2, chemical structure $C_{42}H_{68}O_{13}$, molecular weight = 781 g/mol, purity = 99.5%, BD150742) were purchased from bidepharm(Shanghai,China). Additionally, the following reagents were purchased:Oxidized low-density lipoprotein (ox-LDL, YB-002, Yiyuan Biotech, China), Rapamycin (HY-10219, MCE, USA), Ultra-sensitive ECL chemiluminescence kit (M2301, HaiGene, China), Reactive Oxygen Species Assay Kit(S0033S, Beyotime, China),Oil Red O (O8010, Solarbio, China), total cholesterol assay kit (A111-1–1, njjcnio, China), triglyceride assay kit (A110-1–1, njjcbio,

China), high-density lipoprotein cholesterol assay kit (A112-1–1, njjcbio, China), low-density lipoprotein cholesterol assay kit (A113-1–1, njjcbio, China), creatinine assay kit (C011-2–1, njjcbio, China), aspartate aminotransferase assay kit (C010-2–1, njjcbio, China), alanine aminotransferase assay kit (C009-2–1, njjcbio, China).

## 2.2 Animal programs

30 Male ApoE$^{-/-}$ mice at 7 weeks of age were obtained from the Animal Experimental Center of North Sichuan Medical College (Sichuan, China). All intervention measures and animal care protocols were conducted in accordance with the Animal Ethics Committee guidelines provided by the North Sichuan Medical College (NSMC2024100, Sichuan, China). The mice were housed in temperature-controlled facilities (temperature: 25±2°C, humidity: 55±5%), with a 12-hour light-dark cycle, which continued until the completion of sample collection [26]. They were acclimatized for 1 week with ad libitum access to standard maintenance feed and sterile drinking water before experimental procedures commenced.

At 8 weeks of age, ApoE$^{-/-}$ mice were randomly divided into 5 groups (n = 6 per group). Due to the poor water solubility of the administered compounds, a vehicle was prepared with the following composition: 10% DMSO (HY-Y0873, MCE, USA), 40% PEG300 (HY-Y0873, MCE, USA), 5% Tween-80 (HY-Y1891, MCE, USA), and 45% saline solution [27,28]. The groups included control group (Con,vehicle,ip.), model group (Mod,vehicle,ip.), NGR1 group (NGR1,30 mg/kg/day, ip.), SSB2 group (SSB2,30 mg/kg/day, ip.), and NGR1 + SSB2 group (NS,NGR1 30 mg/kg/day + SSB2 30 mg/kg/day, ip.). The dosage of 30 mg/kg/day was selected based on previous pharmacological studies indicating its effectiveness in mitigating atherosclerotic changes while minimizing potential toxicity [29,30]. All groups, except the control group, were fed a Western diet containing 21% fat and 0.15% cholesterol for 12 weeks. Starting from the 9th week, mice received intraperitoneal injections daily for 4 weeks. For intraperitoneal injections, mice were briefly anesthetized with 2% isoflurane to minimize distress. Mice were monitored daily for signs of pain or distress, and body weight was recorded weekly [26]. No unexpected adverse events occurred. After the completion of the task, mice were euthanized via cervical dislocation under deep anesthesia induced by intraperitoneal injection of sodium pentobarbital (50 mg/kg) [31].

## 2.3 Histological analysis

Liver tissues were fixed in 10% formalin for 24 hours, then embedded in paraffin or OTC, and consecutively sectioned at a thickness of 7µm. Standard hematoxylin and eosin (H&E) staining was performed on paraffin-embedded tissue sections, while frozen sections of liver embedded in OTC were stained with Oil Red O (ORO) to assess hepatic lipid accumulation and tissue damage.

The aorta was dissected from the heart base of the mice and fixed in 4% paraformaldehyde. The aortic tissues were embedded in OCT and sectioned at a thickness of 7 µm. Subsequently, sections of the aortic sinus were stained with hematoxylin and eosin (H&E) and Oil Red O (ORO). Additionally, the entire aorta was longitudinally sectioned and stained with Oil Red O to visualize lipid deposition. The areas of lesion in the aorta and aortic sinus were quantified using ImageJ software [32].

## 2.4 Immunofluorescence staining

The aortic roots of mice or the treated cells were fixed with 4% paraformaldehyde. After permeabilization with Triton X-100, the samples were blocked with BSA and subsequently incubated overnight at 4°C with P62 antibody (M00300-1, Boster, China) at a dilution of 1:100, p-mTOR antibody (67778–1-Ig, Proteintech, China) at a dilution of 1:100, and LAMP-1 (CL647–65050, Proteintech, China) at a dilution of 1:100. Following incubation with FITC-conjugated antibodies and DAPI staining, the cells were washed with PBS and mounted using an anti-fade reagent. Observations were conducted using an inverted fluorescence microscope.

## 2.4 Biochemical Analysis

Blood was collected from fasting mice via retro-orbital puncture into Eppendorf tubes pre-coated with heparin, and centrifuged at 5000 rpm for 10 minutes at 4°C to isolate plasma. Hepatic and renal indicators, as well as lipid markers, were measured in the plasma samples according to the manufacturer's instructions. These markers included alanine aminotransferase (ALT)(C009-2–1, NJJCBIO, china), aspartate aminotransferase (AST)(C010-2–1, NJJCBIO, china), creatinine (Cr)(C011-2–1, NJJCBIO, china), total cholesterol (TC)(A111-1–1, NJJCBIO, china), triglycerides (TG)(A110-1–1, NJJCBIO, china), low-density lipoprotein cholesterol (LDL-C)(A113-1–1, NJJCBIO, china), and high-density lipoprotein cholesterol (HDL-C)(A112-1–1, NJJCBIO, china).

## 2.5 Network pharmacology

Potential targets of NGR1 and SSB2 were identified through SwissTargetPrediction [33](http://www.swisstargetprediction.ch/), Encyclopedia of Traditional Chinese Medicine [34] (http://www.tcmip.cn/ETCM/index.php/Home/Index/All), Traditional Chinese Medicine Systems Pharmacology Database and Analysis Platform [35](https://old.tcmsp-e.com/tcmsp.php), and Herb [36](http://herb.ac.cn/). Molecular targets related to atherosclerosis were screened using GeneCards [37](https://www.genecards.org/), PharmGKB [38](https://www.pharmgkb.org/), and OMIM [39](https://omim.org/) databases. A protein-protein interaction (PPI) network was constructed using STRING 11.0 database [40](https://string-db.org) and visualized using Cytoscape 3.9.1 to identify key targets based on degree and Maximum Clique Centrality (MCC). Subsequently, key shared targets in the PPI network for NGR1-SSB2-atherosclerosis were imported into R software for pathways enrichment analysis of Gene Ontology (GO) and Kyoto Encyclopedia of Genes and Genomes (KEGG) using 'enrichplot' and 'clusterProfiler' R packages [41].

## 2.6 Cell culture

RAW264.7 cells (CL-0190) were purchased from Procell(china,wuhan). The cells were cultured in Dulbecco's Modified Eagle Medium (DMEM; KGL1206−500, KeyGEN bioTECH, China), supplemented with 10% fetal bovine serum from South America (FBS,10270106, Gibco, USA)at 37°C in a humidified atmosphere containing 5% $CO_2$.

## 2.7 CCK8 assay

Cell proliferation was assessed using the CCK-8 kit (KGA9305−500, KeyGEN BioTECH, China). A total of $6 \times 10^3$ cells per well were seeded in a 96-well plate. The cells were stimulated with varying concentrations of ox-LDL, NGR1, and SSB2, and then incubated under appropriate conditions for 24 hours. Afterward, the culture medium was removed and DMEM containing 10% CCK-8 solution was added to each well. The plates were incubated for an additional 2 hours Absorbance was measured at 450nm using a microplate reader (Bio-Tek, Winooski, VT) after shaking for 30 seconds.

## 2.8 Oil Red O staining and dil labeling

Oil Red O staining was employed to observe changes in lipid droplet phagocytosis in RAW264.7 cells. A total of $4 \times 10^{43}$ cells per well were seeded in a 24-well plate with coverslips. The RAW264.7 cells were treated with ox-LDL, NGR1, and SSB2 for 24 hours. Then, the cells were gently washed with PBS and fixed with 4% paraformaldehyde solution for 3 minutes. They were then stained with freshly prepared Oil Red O working solution in a 37°C water bath for 15 minutes. Followed by counterstain with hematoxylin for 5 minutes. Staining and cell morphology were observed under an optical microscope (OLYMPUS, Japan),where lipid droplets within the cells appear as red granules, and cell nuclei as blue and elliptical.

Similarly, after fixing RAW264.7 cells, DiI labeling was performed by adding the Dil working solution directly to the cells. The cells were stained for 10 minutes and then observed under an inverted fluorescence microscope. Lipid droplets exhibited red fluorescence, while cell nuclei showed blue fluorescence.

## 2.9 Analysis of apoptotic cells by flow cytometry

Cell apoptosis was detected using the Annexin V-FITC/PI double staining apoptosis detection kit (KGA1102−50, KeyGEN BioTECH, China). After 24 hours of treatment, RAW264.7 cells were stained with Annexin V-FITC and propidium iodide (PI) in the dark at room temperature. The extent of apoptosis was then analyzed using a flow cytometer (Agilent, USA).

## 2.10 Measurement of cellular reactive oxygen species

RAW264.7 cells were seeded in a 24-well sterile culture plate and treated with the appropriate drug. DCFH-DA (S0033S, Beyotime, China) was diluted with serum-free medium to a final concentration of 10 µmol/L at a ratio of 1:1000. After removing the cell culture medium, the diluted DCFH-DA was added, and the cells were incubated at 37°C for 20 minutes.. The cells were then washed three times with serum-free medium to completely remove any excess DCFH-DA that had not entered the cells. Then, reactive oxygen species were observed using a fluorescence microscope.

## 2.11 RT-qPCR

Total RNA was extracted from macrophages using Trizol reagent (15596018CN, Invitrogen, USA). The extracted RNA was then reverse transcribed into cDNA using a reverse transcription kit (A502, ExonGen, China). qPCR is performed using Fast SYBR Green qPCR Master Mix UDG (A402, ExonGen, China) on the LightCycler® 480 system (Roche LifeScience, USA). Specific primers for tumor necrosis factor alpha (*Tnf-α)*, interleukin-1 Beta *(Il-1β)*, interleukin-18 (*Il-18*), NOD-like receptor family pyrin domain containing 3(*Nlrp3*), peroxisome proliferator-activated receptor γ(*PPAR-γ*), liver X receptor α(*LXR-α)* and ATP-binding cassette transporters A1/G1 *(ABCA1/ABCG1)*were designed using Primer 3 software (Table 1). Expression levels of target gene expression is normalized to β-actin. The relative fold changes in target gene expression were calculated using the $2^{-\Delta\Delta Ct}$ method. Each sample was tested in at least three replicates.

## 2.12 Western blot analysis

Total protein was extracted from cells or tissue using RIPA buffer (BL504A, Biosharp, China) supplemented with protease inhibitors (P1045, Beyotime, China), and phosphatase inhibitors (P1045, Beyotime, China). Protein concentration was measured using a BCA protein assay kit (Bio-Rad, China).The protein samples were then mixed with 5×SDS sample buffer (P0015, Beyotime, China) and heated at 95°C for 15 minutes. Subsequently, 30 µg of protein from each sample was separated by 7%, 10%, or 12.5% SDS-PAGE (60V for 15 minutes; 110V for 75 minutes).The proteins were transferredl to a 0.22µm PVDF membrane (Millipore, USA) (200mA for 80 minutes). The membrane was blockedwith 5% non-fat milk in TBST and incubated with diluted primary antibodies at 4°C overnight on a shaker. The antibodies included: AKT Monoclonal antibody (60203–2-Ig, Proteintech, China), Phospho-AKT (Ser473) Monoclonal antibody (66444–1-Ig, Proteintech,

**Table 1. Primers used for RT-qPCR.**

| Primer Name | Forward Primer (5'-3') | Reverse primer (5'-3') |
|---|---|---|
| *β-actin* | CTGGCTCCTAGCACCATGAAGA | ACAGTCCGCCTAGAAGCACTTG |
| *TNF-α* | CCCTCCAGAAAAGACACCATG | CACCCCGAAGTTCAGTAGACAG |
| *IL-1β* | GAAATGCCACCTTTTGACAGTG | TGGATGCTCTCATCAGGACAG |
| *IL-18* | GTGAACCCCAGACCAGACTG | CCTGGAACACGTTTCTGAAAGA |
| *Nlrp3* | GATCTTCGCTGCGATCAACA | GGGATTCGAAACACGTGCATTA |
| *PPAR-γ* | CATTCTGGCCCACCAACTTC | TCAAAGGAATGCGAGTGGTCTT |
| *LXR-α* | CCTTCCTCAAGGACTTCAGTTACAA | CATGGCTCTGGAGAACTCAAAGAT |
| *ABCA1* | GCGGACCTCCTGGGTGTT | CAAGAATCTCCGGGCTTTAGG |
| *ABCG1* | AAGGCCTACTACCTGGCAAAGA | GCAGTAGGCCACAGGGAACA |

China), mTOR Monoclonal antibody (66888–1-Ig), Phospho-mTOR (Ser2448) Monoclonal antibody (67778–1-Ig, Proteintech, China), PI3 Kinase p110 Beta Polyclonal antibody (20584–1-AP, Proteintech, China), Phospho-PI3 Kinase p85 (Tyr458)/p55 (Tyr199) Antibody (4228T, CST, USA), IL-1 Beta Polyclonal antibody (16806–1-AP, Proteintech, China), IL-18 Polyclonal antibody (10663–1-AP, Proteintech, China), Nlrp3 Polyclonal antibody (30109–1-AP, Proteintech, China), Caspase 3/p17/p19 Polyclonal antibody (19677–1-AP, Proteintech, China), BAX Polyclonal antibody (50599–2-Ig, Proteintech, China), Bcl2 Polyclonal antibody (A00040-1, Boster, China), Anti-LC3B/MAP1LC3B Antibody (BM4827, Boster, China), Beclin-1 Antibody (3738T, CST, USA), Anti-P62/SQSTM1 Antibody (M00300-1, Boster, China).After incubation with secondary antibodies (BA1056, Boster, China) for 1 hour, protein bands were visualized using an enhanced ECL chemiluminescence reagent (P0018AM, Beyotime, China) and an Odyssey infrared imaging scanner (LI-COR Biosciences). The intensity of the bands was analyzed using Image-J software (NIH, Bethesda, MD) and and protein expression levels were normalized to β-actin as a loading control.

### 2.13 Statistical analysis

All data were presented as the mean±standard deviation (SD). Statistical analysis were performed using GraphPad Prism 8.0 (GraphPad Software, Inc.). Comparisons between multiple groups were made using one-way analysis of variance (ANOVA), with *p*-values<0.05 considered statistically significant.

## 3 Results

### 3.1 Network pharmacology analysis reveals a close association between NS and atherosclerosis

To explore the potential mechanisms of NS in combating atherosclerosis, we employed network pharmacology approaches A total of 241 NGR1-related targets, 222 SSB2-related targets, and 4619 atherosclerosis-related targets were identified, with113 shared targets (Fig 1A). Enrichment analysis using KEGG and GO indicated that NS may modulate processes involved in inflammation, apoptosis, autophagy, and the PI3K/AKT signaling pathway to mediate the progression of atherosclerosis (Fig 1B-C). The protein-protein interaction network comprised 110 nodes and 1732 edges (Fig 1D). Through target contribution (degree) and maximum clique centrality (MCC), 10 key proteins including CASP3 and MTOR were identified as core targets (Fig 1E, 1F).

### 3.2 NS promotes lipid transport in the ApoE⁻/⁻ mouse model

The atherosclerosis model was induced in ApoE$^{-/-}$ mice by feeding Western diet for 12 weeks. During this period, the control group showed a declining trend in body weight from weeks 8–12 (Fig 2A). In contrast, the other groups did not show significant differences in weight change during this time.To investigate the effect of the NS combination on lipid metabolism *in vivo*, we measured four blood lipid parameters in these ApoE$^{-/-}$ mice fed the Western diet. Compared to the control group, the model group showed significantly elevated levels of TC, TG, and LDL-C, along with a significant reduction in HDL-C. Treatment with NGR1, SSB2, or the NS combinationreduceed the levels of TC, TG, and LDL-C levels,while increaseing HDL-C levels in ApoE$^{-/-}$ mice. Among them, the NS combination group showed the most significant effects (Figs 2B–E). Furthermore, we conducted biochemical and morphological analyses of the liver and kidneys in these mice. The Western diet increased serum ALT and AST levels in the liver. Compared to the model group and the monotherapy group, the NS combination therapy group showed reduced serum ALT and AST levels (Figs 2F-G). H&E staining of liver sections revealed significant hepatocellular steatosis induced by the Western diet, which was mitigated under NS treatment, as evidenced by reduced lipid vacuoles and inflammatory changes. Oil Red O staining of the liver confirmed a notable decrease in lipid droplets in the NS-treated group (Fig 2I). There were no significant differences observed among groups in kidney morphology (S1 Fig).

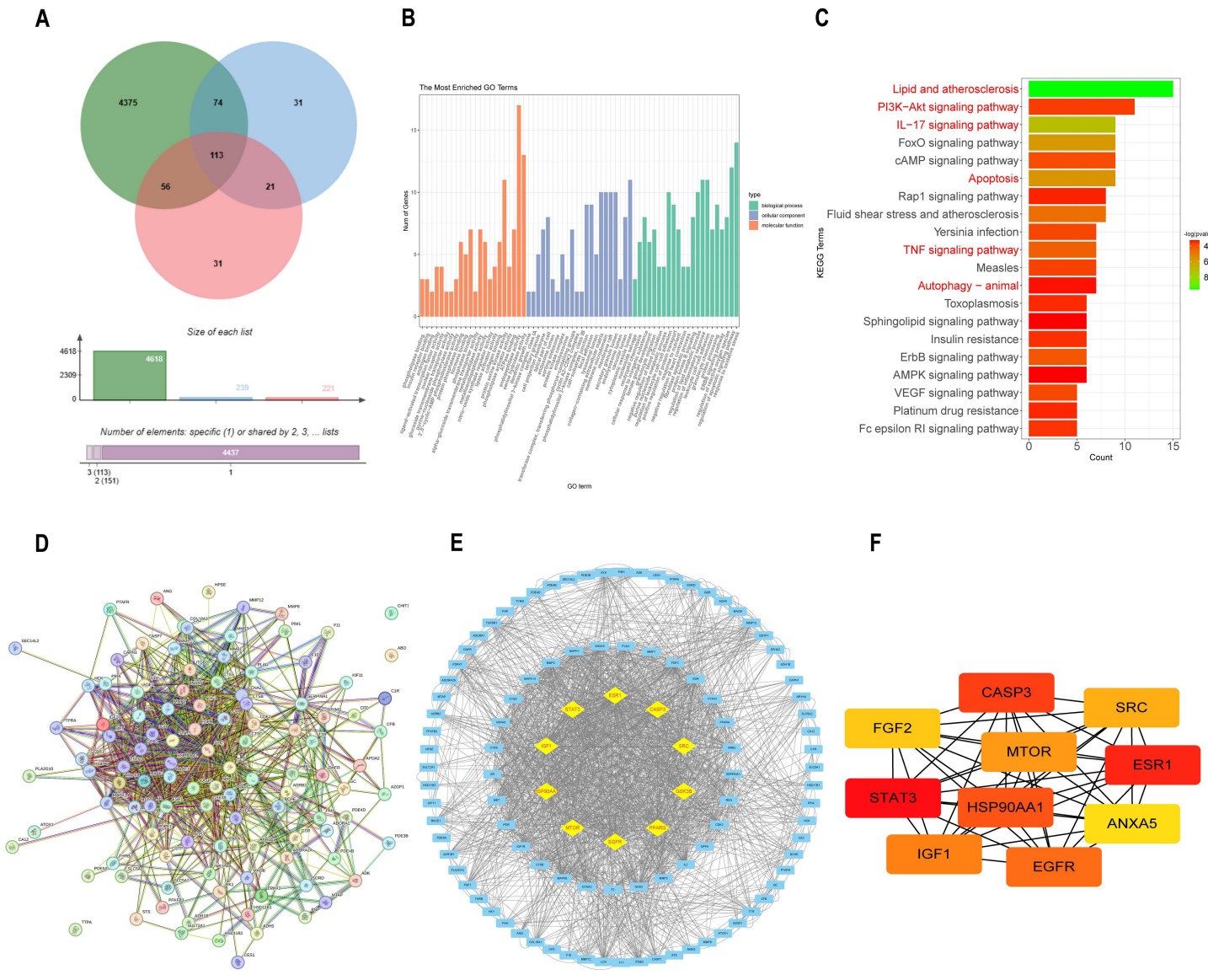

**Fig 1. Network pharmacology analysis predicted the targets of NS on atherosclerosis.** (A) Venn analysis of NGR1, SSB2, and atherosclerosis overlapping targets.(B) GO enrichment analysis of NS-atherosclerosis shared targets.(C) KEGG enrichment analysis of NS-atherosclerosis shared targets.(D) PPI (Protein-Protein Interaction) network of NS-atherosclerosis shared targets.(E) Top 10 core targets selected from a list of 113 targets based on degree contribution.(F) Identification of 10 targets from shared targets based on Maximum Clique Centrality (MCC).

### 3.3 NS delays the formation of atherosclerotic plaques in the ApoE$^{-/-}$ mouse model

To investigate the effects of NS on atherosclerotic plaques, we assessed the changes in the entire aortic tissues of high-fat-fed ApoE$^{-/-}$ mouse models. Oil Red O staining revealed that the stained area in the NS-treated group was smaller compared to both the high-fat diet group and the monotherapy groups (Fig 3C). Histological examination of aortic root sections under optical microscopy showed severe lipid accumulation, and the formation of unstable plaques with necrotic cores in the model group (Figs 3A-B). In contrast, the plaque area was reduced in the NGR1 and SSB2 treatment groups. Notably, the NS combination intervention significantly limited necrotic area and reduced lipid deposition

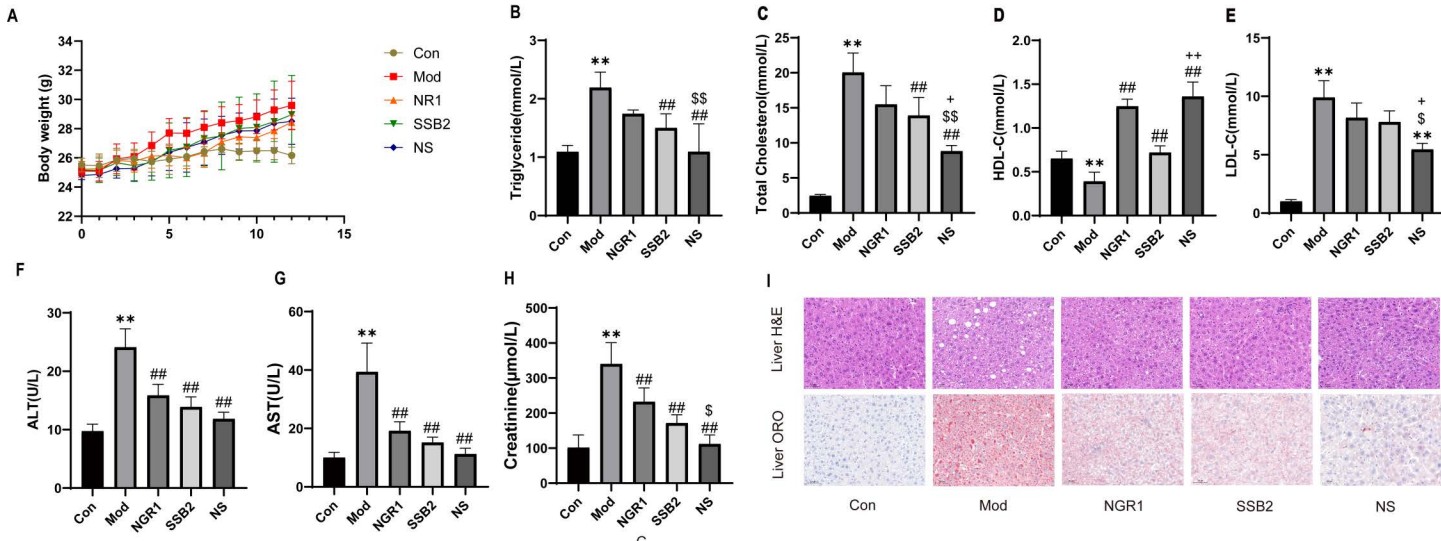

**Fig 2. NS affects lipid transport in ApoE⁻/⁻ mouse models.** Eight-week-old male ApoE-/- mice, weighing approximately 25g, were fed a Western high-fat diet for 12 weeks. They were then intraperitoneally injected with NGR1 (30 mg/kg/day) and SSB2 (30 mg/kg/day) for 4 weeks. (A) Effects of NGR1 and SSB2 on the body weight of ApoE-/- mice. (B) Triglycerides (TG), (C) total cholesterol (TC), (D) high-density lipoprotein cholesterol (HDL-C), (E) low-density lipoprotein cholesterol (LDL-C), (F) alanine transaminase, (G) aspartate transaminase, and (H) creatinine levels in the control group(Con), model group(Mod), NGR1 group(NGR1), SSB2 group(SSB2), and NGR1 + SSB2 combination treatment group(NS). (I) Representative images of liver sections stained with H&E and Oil Red O. Tissues were examined using an optical microscope (scale bar = 50 μm).All data were presented as the mean ± SD. **$p < 0.01$ vs. control grou*p*, ##$p < 0.01$ vs. model grou*p*,$$p < 0.05$, $$$p < 0.01$ vs. NGR1 grou*p*, +$p < 0.05$,++$p < 0.01$ vs. SSB2 grou*p*.

to stabilize aortic plaques.Macrophage accumulation, a hallmark of atherosclerotic plaques, was assessed using CD68 immunofluorescence analysis. The results indicated that NS treatment significantly reduced the accumulation of macrophages in atherosclerotic lesions.

### 3.4 NS combination attenuates ox-LDL-induced foam cell formation

Before assessing the *in vitro* effects of two traditional Chinese medicine monomers, RAW 264.7 macrophage cells were exposed for 24 hours to various concentrations of NGR1 and SSB2, and their cytotoxicity was evaluated using the CCK-8 assay. As shown in Fig 4A, SSB2 did not demonstrate significant cytotoxic effects at concentrations ranging from 12.5 to 50 μM. However, at 100 μM (cell viability 84.09%) and 200 μM (cell viability 61.94%), it exhibited notable cytotoxicity, leading to the selection of 50 μM for further experiments. Meanwhile, NGR1 did not display significant cytotoxicity across concentrations from 12.5 to 200 μM, with a trend of increased cell viability at lower concentrations, peaking at 50 μM (Fig 4B). Therefore, the concentration of 50 μM was chosenas the optimal concentration for NGR1 as well.When over-loaded with ox-LDL, macrophages uptake excessive lipids, inducing the formation of foam cells, and ultimately leading to the instability and rupture of atherosclerotic plaques.To explore the effects of NS on foam cell formation induced by ox-LDL, we established a foam cell model by treating macrophages with 50 μg/ml ox-LDL. We found that after incubation with ox-LDL, RAW 264.7 macrophages transitioned from a round shape to an irregular morphology with pseudopodia, accompanied by significant cytoplasmic lipid droplet accumulation. In contrast, treatment with NGR1 or SSB2 corrected cell morphology and lipid droplet burden. Compared to the individual treatment groups, combination therapy markedly reduced lipid droplet accumulation (Figs 4C-D). In order to further investigate the regulatory mechanisms by which NS modulates lipid transport, we assessed the expression of mRNA associated with cholesterol transport. The results indicated that ox-LDL stimulation significantly downregulated the expression levels of PPAR-γ, LXR-α, ABCA1, and ABCG1(Figs 4E-H). Notably,

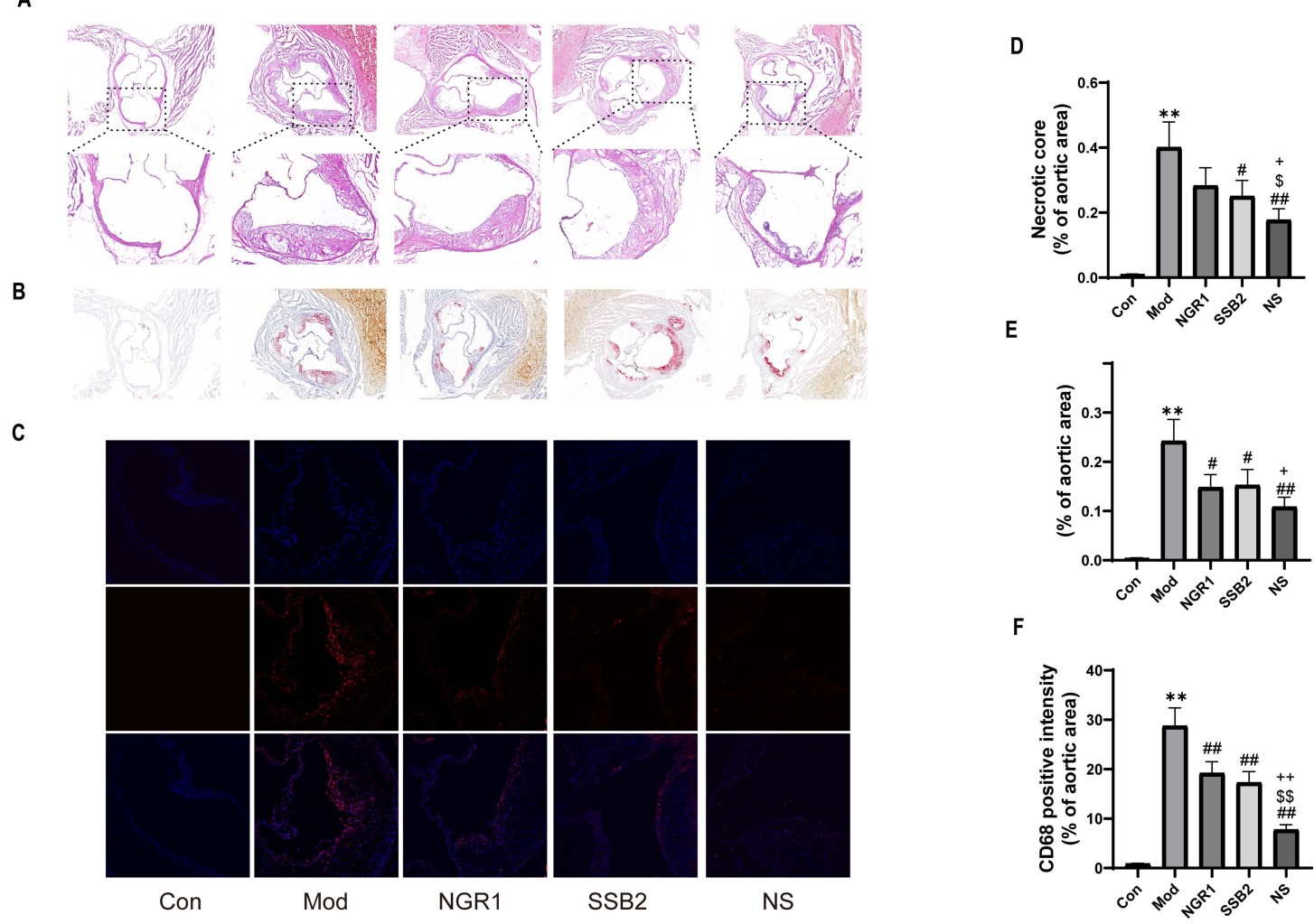

**Fig 3. NS delays the formation of atherosclerotic plaques in ApoE⁻ᐟ⁻ mouse models.** (A, D) Representative images and quantitative analysis of the aortic roots from ApoE⁻ᐟ⁻ mice using H&E staining (scale bars = 200 μm and 100 μm, respectively). (B, E) Representative images and quantitative analysis of the aortic roots from ApoE⁻ᐟ⁻ mice using Oil Red O staining (scale bar = 200 μm). (C, F) Representative images and quantitative analysis of CD68 immunostaining in the aortic root (scale bar = 50 μm).All data were presented as the mean ± SD, (n = 3). **$p < 0.01$ vs. control group*p*, #$p < 0.05$, ##$p < 0.01$ vs. model group, \$$p < 0.05$, \$\$$p < 0.01$ vs. NGR1 group*p*, +$p < 0.05$, ++$p < 0.01$ vs. SSB2 grou*p*.

NS was able to restore the expression levels of these key factors, demonstrating a more pronounced effect compared to the monotherapy group. These findings suggest that NS may collaboratively inhibit ox-LDL-induced lipid transport in RAW 264.7 macrophages, thereby impeding the process of foam cell formation.

### 3.5 NS inhibits inflammation induced by Ox-LDL in RAW264.7 cells

Based on *in vivo* experiments and network pharmacology results, we further investigated whether NS could inhibit ox-LDL-induced macrophage inflammatory responses. Results from Fig 5(A-D) revealed that NR1 and SSB2 significantly reduced the gene expression levels of inflammatory cytokines (TNF-α, IL-1β, IL-18) as well as NLRP3. Visual and quantitative results from Western blotting indicated that ox-LDL stimulation markedly increased the expression of NLRP3

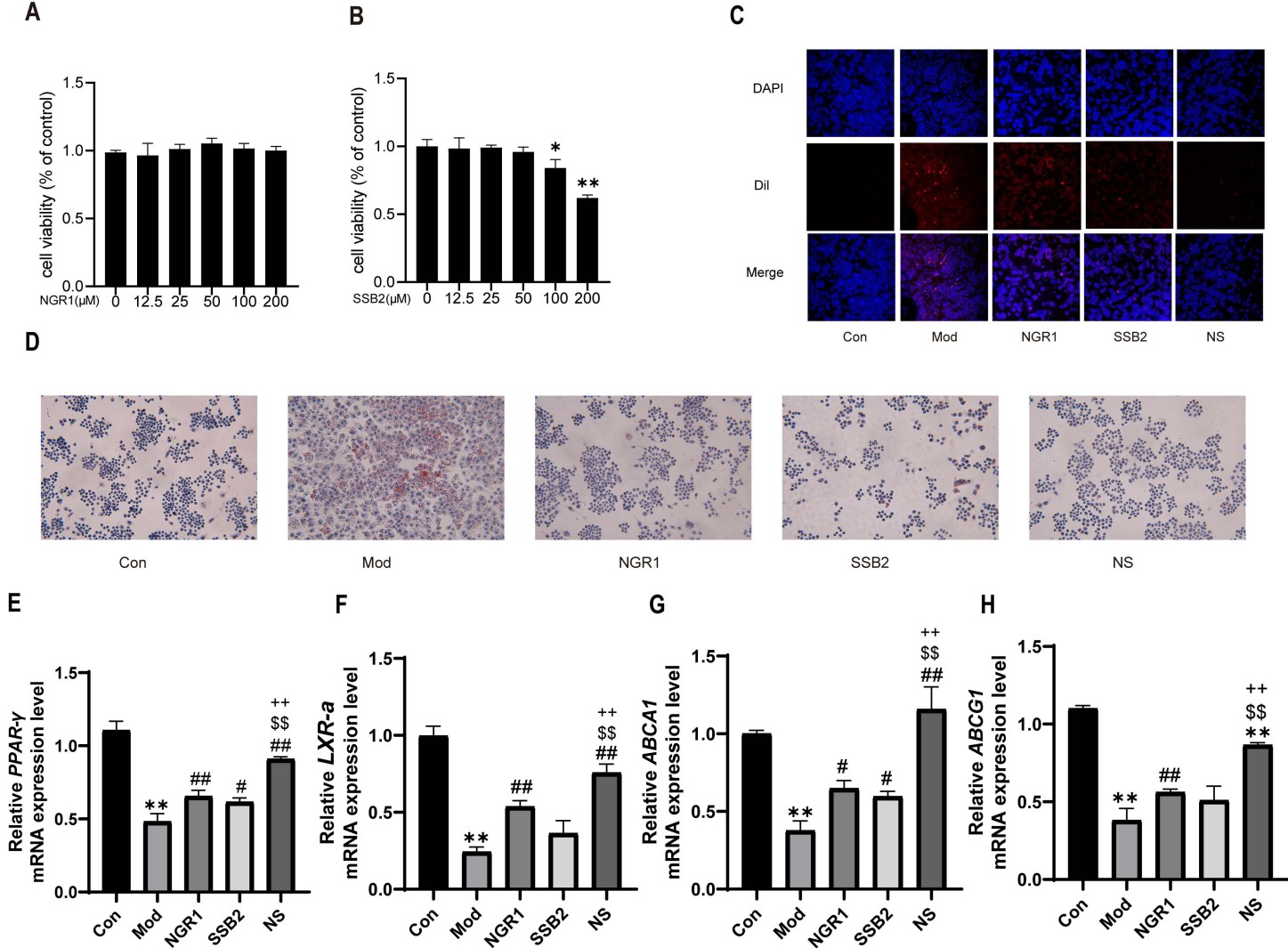

**Fig 4. NS attenuates ox-LDL-induced foam cell formation.** (A) Effect of NGR1 on RAW264.7 cell viability. (B) Effect of SSB2 on RAW264.7 cell viability.(C) Representative fluorescence images of RAW264.7 cells labeled with Dil-oxLDL. Cells were visualized using inverted fluorescence microscopy (scale bar = 50 μm). (D) Representative images of Oil Red O staining showing lipid droplets accumulation in RAW264.7 cells. Cells were examined using an optical microscope (scale bar = 100 μm). (E-H) The use of gene-specific oligonucleotide primers for the real-time PCR analysis of *PPAR-γ, LXR-α, ABCA1*, and *ABCG1*. All data were presented as the mean ± SD, (n = 3). *$p < 0.05$ vs. control group, **$p < 0.01$, **$p < 0.01$ vs. control group, #$p < 0.05$, ##$p < 0.01$ vs. model group, $$$p < 0.01$ vs. NGR1 group, ++$p < 0.01$ vs. SSB2 group.

inflammasome-related proteins, including NLRP3, IL-1β, and IL-18.Treatment with NGR1 or SSB2 treatment slightly reversed this upregulation, with the combination therapy showing a more pronounced inhibitory effect (Figs 5E-H).

### 3.6 NS inhibits apoptosis induced by Ox-LDL in RAW264.7 cells

Network pharmacology results indicate that the anti-atherosclerotic effects of NS are closely associated with apoptosis, with CASP3 playing a crucial role in apoptosis and identified as a potential key target of NS in its anti-atherosclerotic action. To understand the role of NS in ox-LDL-induced apoptosis in RAW264.7 cells, we measured intracellular ROS levels and assessed apoptosis using flow cytometry and western blotting.Compared to the model group, the NS combination

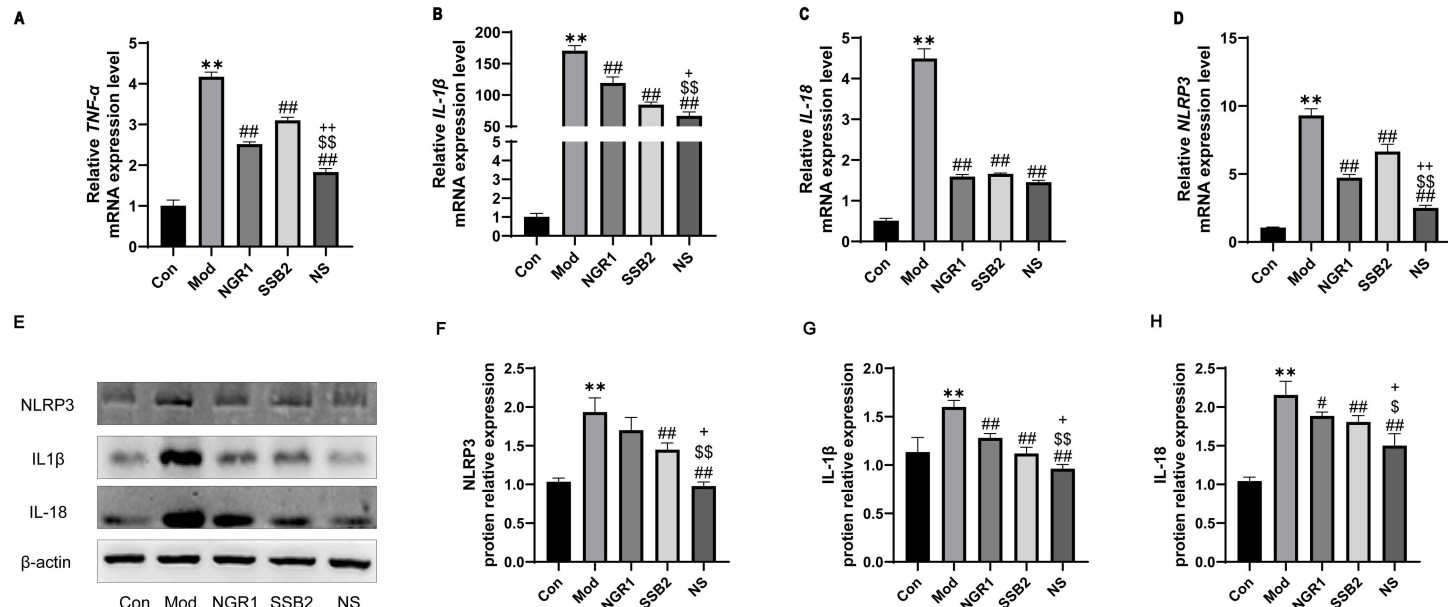

**Fig 5. NS suppresses inflammation induced by Ox-LDL in RAW264.7 cells.** Relative mRNA levels of inflammatory genes. (A) *TNF-α*,(B) *IL-1β*, (C) *IL-18*, and (D) NLRP3 measured by RT-qPCR. (E) Representative Western blot images showing protein expression of NLRP3, IL-1β, and IL-18 in the Ox-LDL-induced cell model. β-actin was used as a loading control. (F-H) Quantification of the relative protein levels of NLRP3, IL-1β, and IL-18, normalized to β-actin and presented as relative intensity based on Western blot.analysis. All data were presented as the mean±SD, (n=3). **$p<0.01$ vs. control group, #$p<0.05$, ##$p<0.01$ vs. model group, \$$p<0.05$, \$\$$p<0.01$ vs. NGR1 group, +$p<0.05$, ++$p<0.01$ vs. SSB2 group.

treatment significantly reduced the ROS levels in RAW264.7 cells (Fig 6A). As shown in Fig 6B, compared to the control group, ox-LDL-treated RAW264.7 cells exhibited a significantly increased number of apoptotic cells, this increase was reversed by NS treatment. Consistent with the changes in apoptosis data, ox-LDL induction in the model group led to the downregulation of the anti-apoptotic protein Bcl-2, while upregulating the pro-apoptotic proteins Bax and cleaved caspase-3. Treatment with either NGR1 or SSB2 attenuated the apoptotic effects of ox-LDL, and compared to single-drug treatment, the combination therapy further inhibited apoptosis in macrophage cells (Figs 6C-F). In summary, these data indicate that NS effectively suppresses ox-LDL-induced apoptosis in RAW264.7 cells.

### 3.7 NS affects the PI3K/AKT/mTOR pathway in RAW264.7 cells

The above results indicate that the PI3K/AKT signaling pathway plays a significant role in the anti-atherosclerotic effect of NS. mTOR is a potential core target for the anti-atherosclerotic effect of NS, acting as a downstream target of the PI3K/AKT pathway and exerting pleiotropic cellular effects.Therefore, we further explored the changes in the PI3K/AKT/mTOR pathway *in vivo* and in ox-LDL-induced macrophages treated with NS. We observed the expression of proteins involved in the PI3K/AKT/mTOR signaling pathway. In ApoE$^{-/-}$ mice, as shown by western blot results, the ratio of p-PI3K/PI3K and p-Akt/Akt in ox-LDL-exposed RAW264.7 cells was significantly higher than in the control group. The monotherapy group did not exhibit significant changes, however, the NS treatment significantly diminished this increase. Similarly, the ratio of p-mTOR/mTOR in ox-LDL-exposed RAW264.7 cells was significantly enhanced compared to the control group. In contrast, NS treatment significantly reduced these ratios (Figs 7A-D). Immunofluorescence analysis further supported these findings. Compared to the control group, the staining intensity of p-mTOR gradually increased in the ox-LDL group. Additionally, p-mTOR expression was reduced in the NS treatment group (Figs 7E-F). These results indicates that NS partially inhibits the activation of the PI3K/Akt/mTOR pathway.

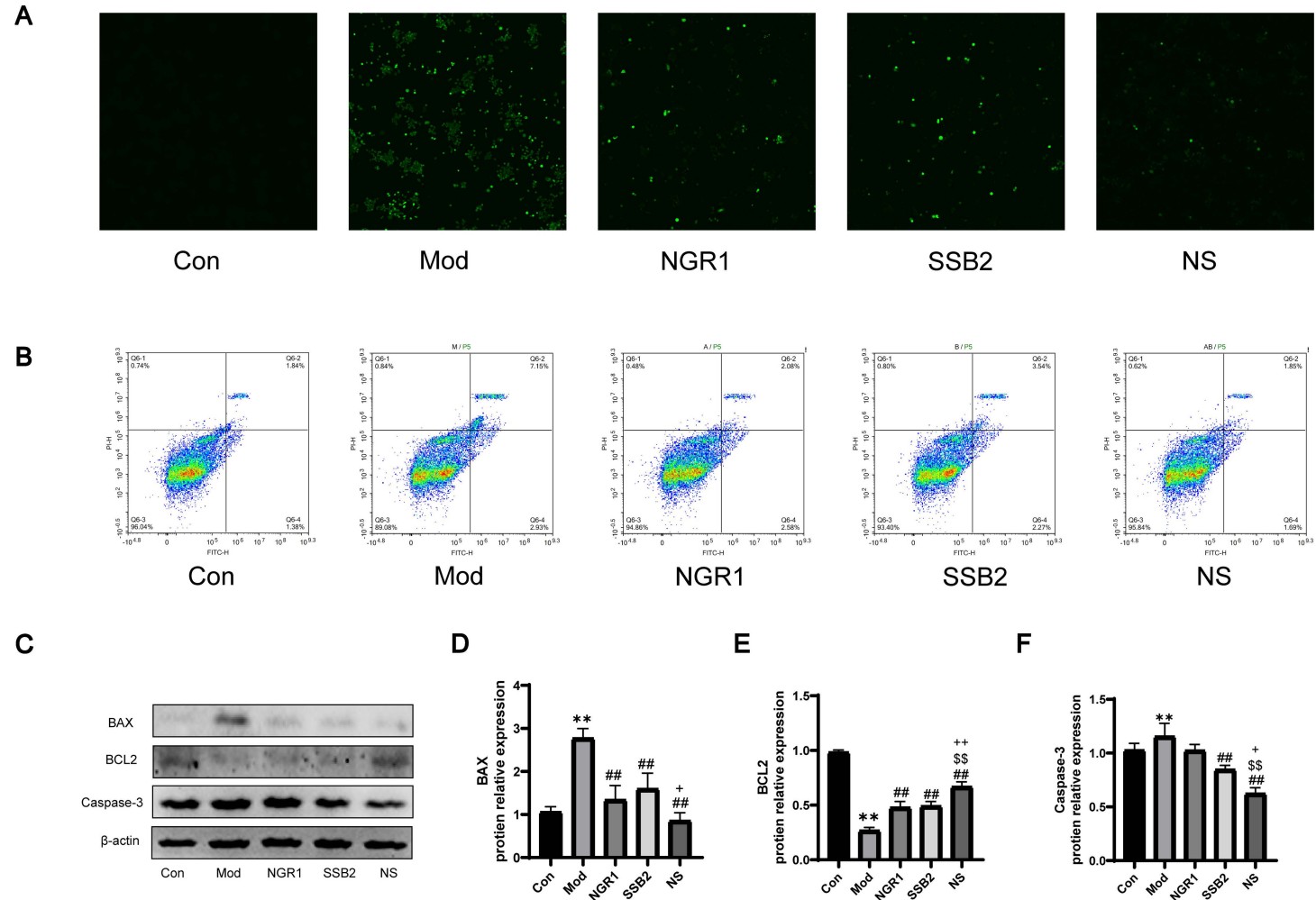

**Fig 6. NS co-suppresses Ox-LDL-induced apoptosis in RAW264.7 cells.** (A) ox-LDL induces ROS generation in RAW264.7 cells, visualized by inverted fluorescence microscopy (scale bar = 50 μm). (B) Representative images of apoptosis detected by flow cytometry. (C) Representative Western blot images showing protein expression of BAX, BCL2, and Caspase-3 in the ox-LDL-induced cell model. β-actin was used as a loading control. (D-F) Quantification of the relative protein levels of BAX, BCL2, and Caspase-3 normalized to β-actin, and presented as relative intensity measured by Western blot analysis. All data were presented as the mean ± SD, (n = 3). ** $p < 0.01$ vs. control group, ## $p < 0.01$ vs. model group, $$ $p < 0.01$ vs. NGR1 group, + $p < 0.05$, ++ $p < 0.01$ vs. SSB2 group.

### 3.8 NS inhibits atherosclerosis by enhancing macrophage autophagic flux

Within the autophagy regulatory pathway, mTOR acts as a negative regulator of autophagy. Thus, inhibiting the PI3K/AKT/mTOR pathway can regulate macrophage autophagy. Using Western blotting, we evaluated the expression of key autophagy proteins Beclin-1, P62, and LC3 in ox-LDL-exposed RAW264.7 cells. Our findings indicated that NS treatment significantly enhances Beclin-1 expression in ox-LDL-exposed RAW264.7 cells. Conversely, NS treatment markedly attenuates P62 expression in ox-LDL-exposed RAW264.7 cells. Additionally, the LC3 II/I protein ratio, which was significantly reduced in ox-LDL-exposed RAW264.7 cells, was significantly reversed by NS treatment. Next, we assessed the expression level of p-mTOR and analyzed its co-localization with lysosomal marker protein LAMP-1 to evaluate lysosomal function through immunofluorescence staining. As shown in Fig 8E, compared with the control group, which displayfewer yellow

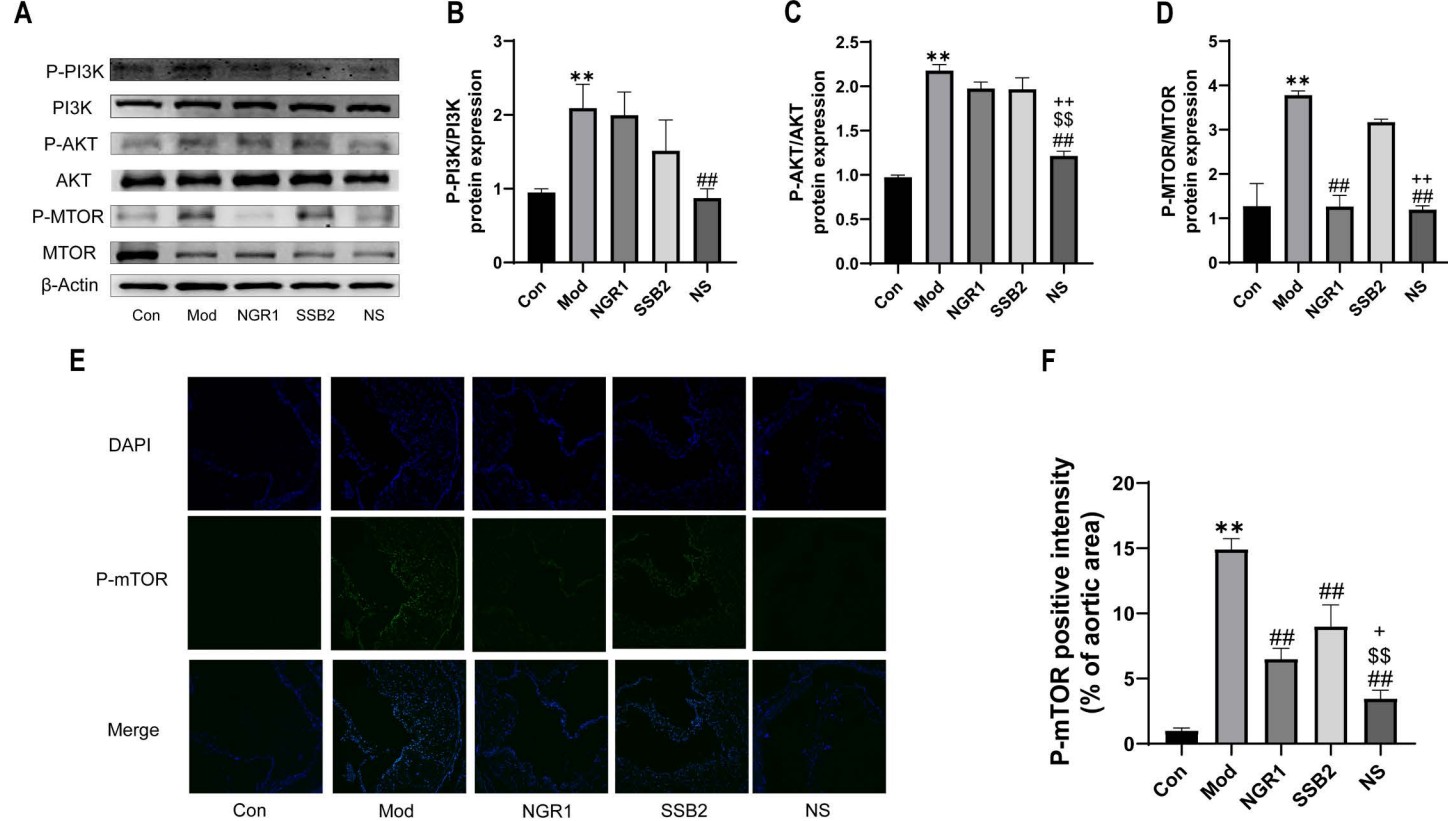

**Fig 7. NS affects the PI3K/AKT/mTOR pathway in RAW264.7 cells.** (A) Representative western blot images showing protein expression of P-PI3K, PI3K, P-AKT, AKT, P-mTOR, and mTOR in the Ox-LDL-induced cell model. β-actin was used as a loading control. (B) Quantitative analysis of P-PI3K/PI3K ratio.(C) Quantitative analysis of P-AKT/AKT ratio. (D) Quantitative analysis of P-mTOR/mTOR ratio. The ratios are presented as relative intensities. (E) Representative fluorescence staining image of P-mTOR examined by inverted fluorescence microscopy (scale bar=50 μm).(F)Quantitative analysis of representative fluorescent staining images of P-AKT, expressed as relative intensity. All data were presented as the mean±SD, (n=3). \*\**p*<0.01 vs. control group, ##*p*<0.01 vs. model group, \$\$*p*<0.01 vs. NGR1 group,+*p*<0.05,++*p*<0.01 vs. SSB2 group.

puncta,ox-LDL-induced RAW264.7 cells exhibited significantly enhanced co-localization of p-mTOR (green) with LAMP-1 (red), with a marked increase in the area of yellow merged puncta in confocal images. Notably, NS treatment significantly suppressed the ox-LDL-induced lysosomal localization of p-mTOR, as evidenced by a substantial reduction in co-localized yellow puncta. *In vivo* Immunofluorescence analysis of P62 to assess autophagic activity in lesions revealed decreased autophagic activity in atherosclerotic lesions(Fig 8F). However, following administration of NS, autophagy significantly recovered (Figs 8F-G). In summary, these results indicate that NS can alleviate atherosclerosis by activating macrophage autophagy.

## 3.9 NS restores autophagy via the PI3K/AKT/mTOR pathway, inhibiting foam cell formation and suppressing inflammation and apoptosis in ox-LDL-stimulated macrophages

To futher determine whether the NS improves atherosclerosis by affecting the PI3K/Akt/mTOR signaling pathway and autophagy, we evaluated the effects of rapamycin (Ra) and 3-MA on ox-LDL-stimulated macrophages. We observed that rapamycin further reduced the accumulation of lipid droplets and the rate of apoptosis induced by ox-LDL. Additionally, it effectively enhanced the transcription levels of *PPAR-γ, LXR-α, ABCA1,* and *ABCG1* genes. In contrast, the beneficial

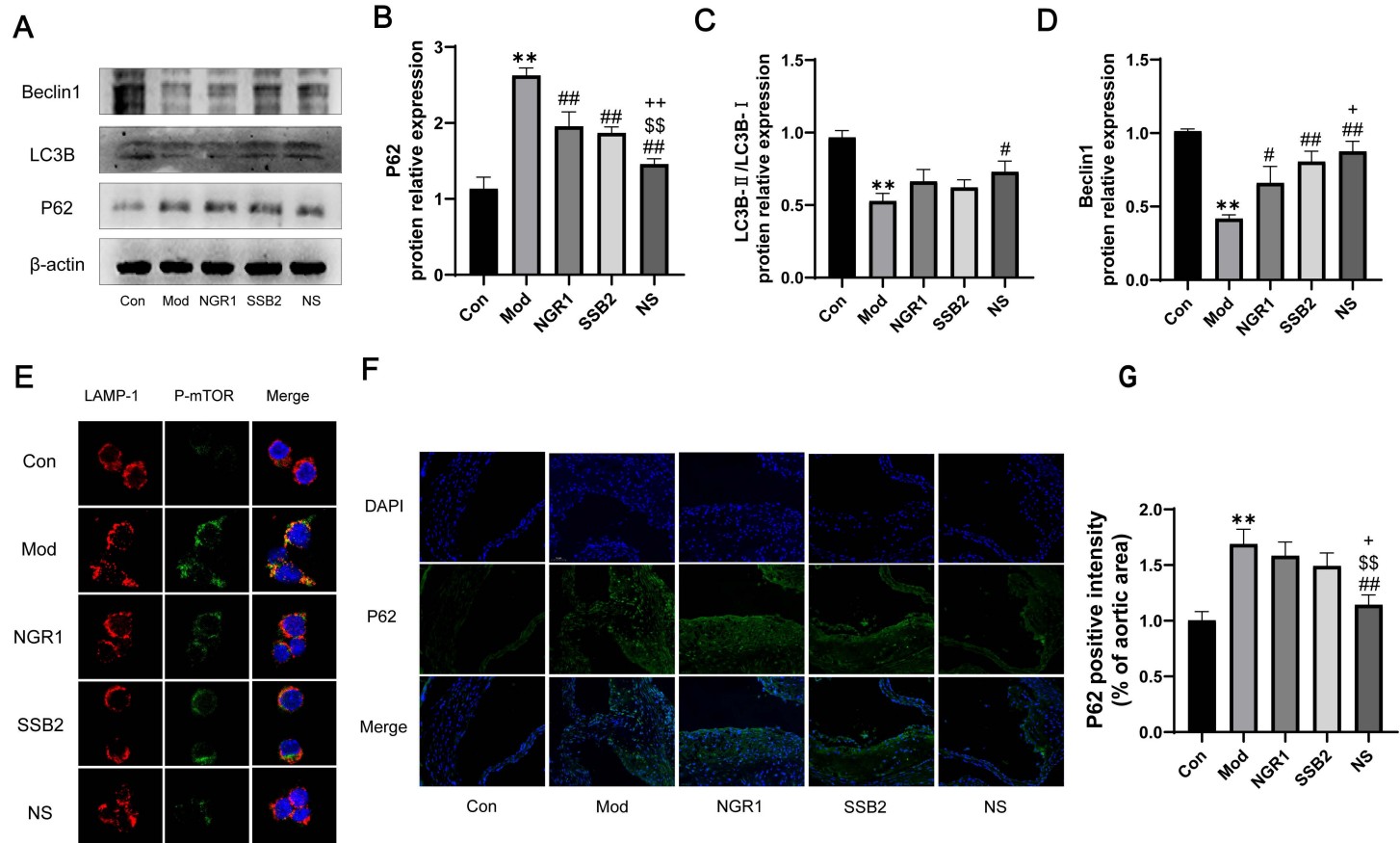

**Fig 8. NS enhances autophagy inhibits atherosclerosis in RAW264.7 cells.** (A) Representative Western blot images showing protein expression levels of BECLIN1, LC3, and P62 in the Ox-LDL-induced cell model. β-actin was used as a loading control. (B-D) Relative protein levels of BECLIN1, LC3, and P62, normalized to β-actin and presented as relative intensities. (E) Representative confocal microscopy images demonstrating the co-localization of p-mTOR (green) and Lamp-1 (red) in RAW264.7 macrophages (scale bar = 10 μm). (F) Representative fluorescence staining images of P62 examined by inverted fluorescence microscopy (scale bar = 50 μm). (G) Quantitative analysis of the fluorescence intensity of P62 from representative staining images. All data were presented as the mean ± SD, (n = 3). **$p < 0.01$ vs. control group, #$p < 0.05$, ##$p < 0.01$ vs. model group, $$$p < 0.01$ vs. NGR1 group, +$p < 0.05$, ++$p < 0.01$ vs. SSB2 group.

effects of NS treatment were partially blocked by 3-MA (Figs 9A-F). Next, we measured the levels of key proteins using Western blot, including NLRP3 (Fig 9H), caspase-3 (Fig 9K), the p-mTOR/mTOR ratio (Fig 9L), and p62 (Fig 9M). Compared to the control group, the Mod group showed a significant increase in these markers. In contrast, Bcl2 (Fig 9J) and Beclin1 (Fig 9N) were significantly reduced in the ox-LDL group, indicating that ox-LDL treatment activates the PI3K/Akt/mTOR pathway and apoptosis while inhibiting autophagic flux and inflammation in RAW264.7 cells. Notably, immuno-fluorescence co-localization analysis revealed that, in comparison to the NS group, rapamycin-treated RAW264.7 cells exhibited further reduced co-localization of p-mTOR with LAMP-1. Conversely, 3-MA-treated RAW264.7 cells demonstrated significantly enhanced p-mTOR/LAMP-1 co-localization, with a marked increase in the area of yellow merged puncta observed in the confocal microscopy images (Fig 9P). As expected, compared to cells treated with ox-LDL alone, NS treatment significantly reduced the expression of NLRP3, p-mTOR, BAX, and p62. The NS treatment group showed significantly higher levels of BCL2, Beclin1, and, LC3-II and LAMP1 compared to the Mod group, with rapamycin further enhancing the effects of NS. However, these effects of NS could be partially eliminated by the autophagy inhibitor 3-MA.

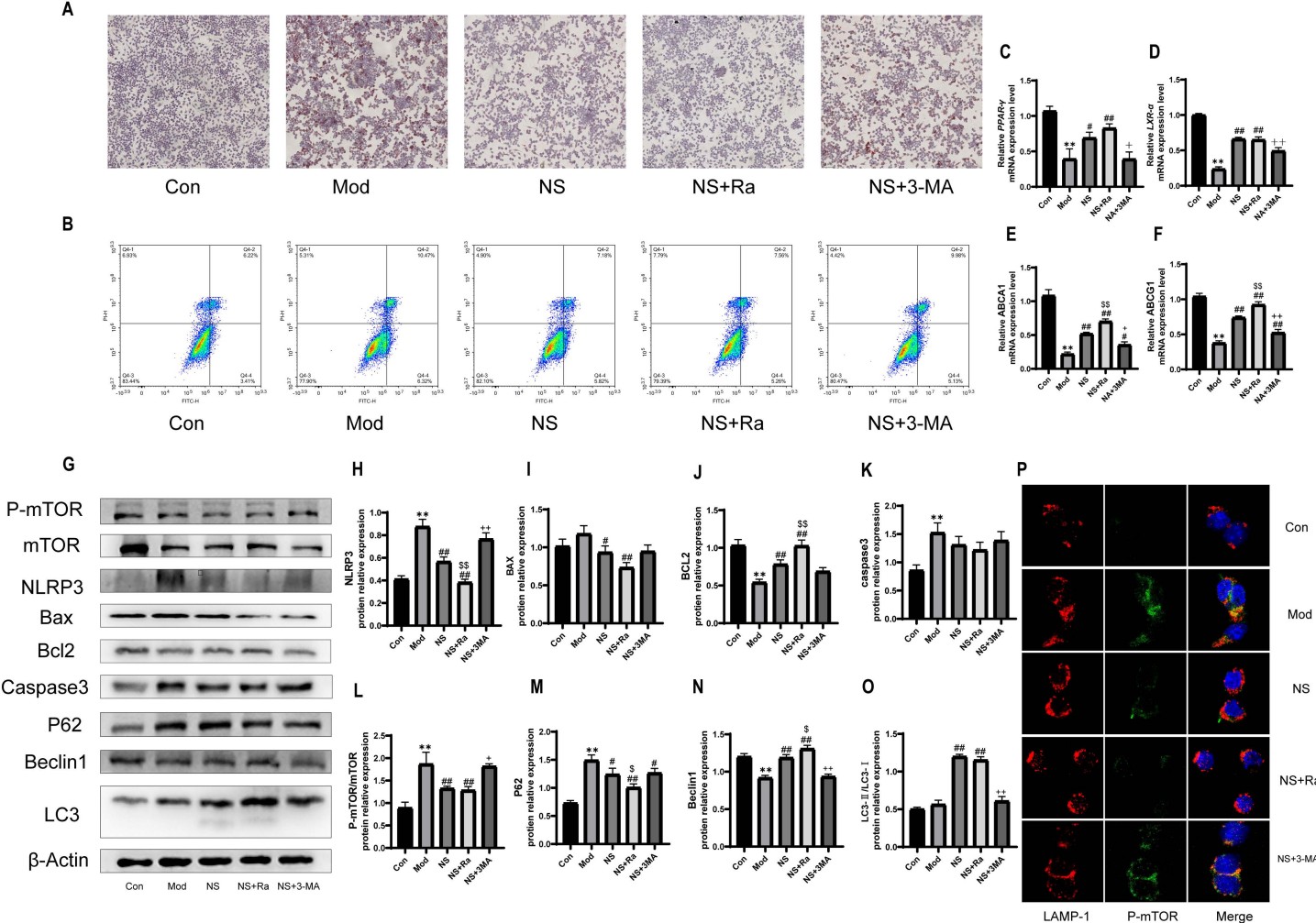

**Fig 9. NS restores macrophage autophagy and alleviates the progression of atherosclerosis by inhibiting the PI3K/Akt/mTOR pathway.** Representative images of Oil Red O staining showing lipid droplets in RAW264.7 cells. Cells were examined under a light microscope (scale bar = 100 μm). (B) Representative images of cell apoptosis detected by flow cytometry. (C-F) The use of gene-specific oligonucleotide primers for the real-time PCR analysis of *PPAR-γ, LXR-α, ABCA1*, and *ABCG1*. (G) Representative Western blot images showing protein expression levels of P-mTOR, mTOR, NLRP3, BAX, BCL2, Caspase3, P62, BECLIN1, and LC3 in ox-LDL-induced cellular models. β-actin was used as the loading control. (H-O) Quantification of relative protein levels were normalized to β-actin and were presented as relative intensity. (P) Representative confocal microscopy images demonstrating the co-localization of p-mTOR (green) and Lamp-1 (red) in RAW264.7 macrophages (scale bar = 10 μm). All data were presented as the mean ± SD, (n = 3). \*\*$p < 0.01$ vs. control group, #$p < 0.05$, ##$p < 0.01$ vs. model group, \$$p < 0.05$, \$\$$p < 0.01$ vs. NS group, +$p < 0.05$, ++$p < 0.01$ vs. NS group. Rapamycin: 0.5 μM, 3-MA: 3 mM.

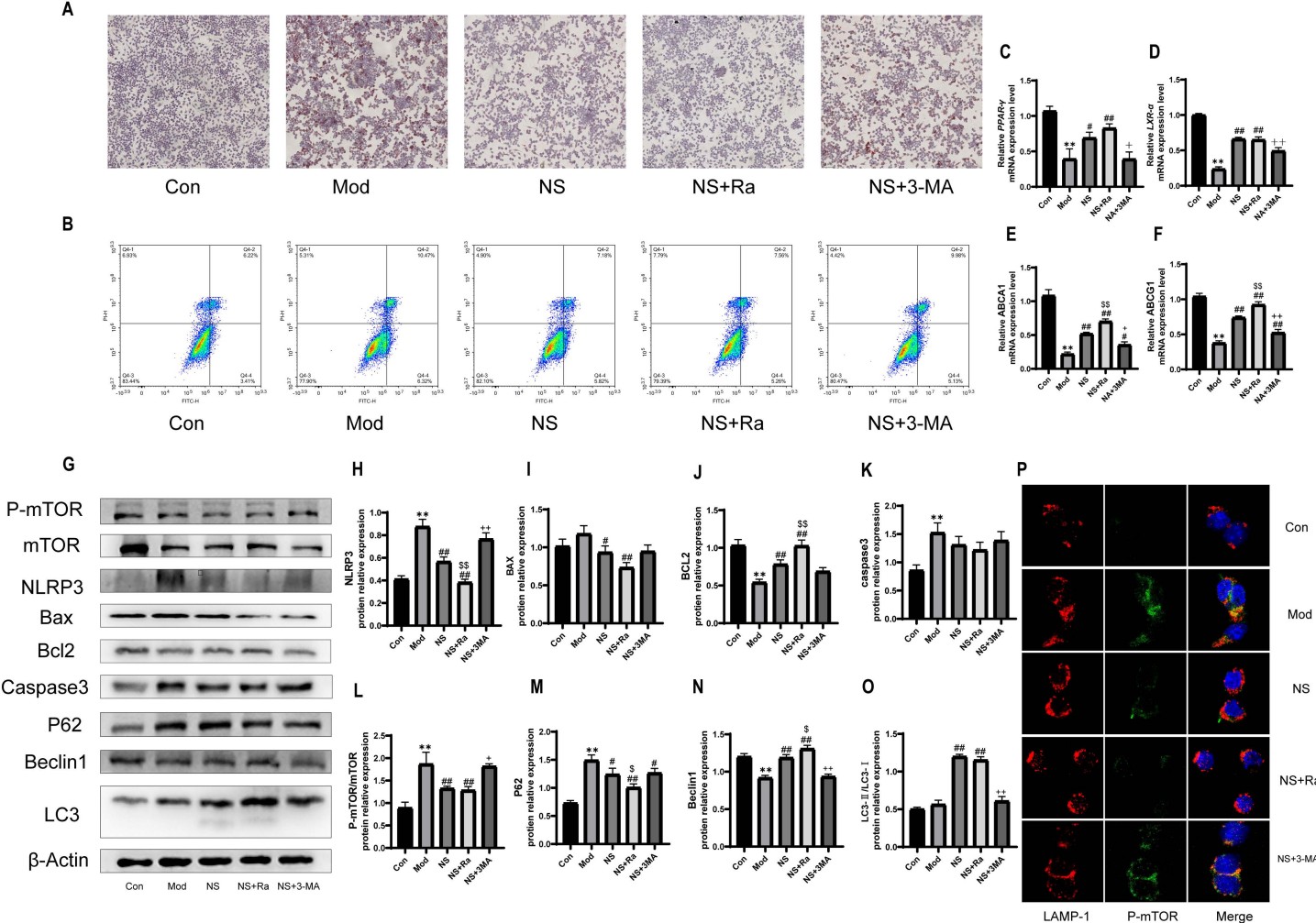

These results suggested that NS mitigates the activation of the PI3K/Akt/mTOR signaling pathway induced by ox-LDL, thereby restoring autophagic flux in ox-LDL-treated macrophages and inhibiting foam cell formation and apoptosis.

## 4 Discussion

Atherosclerosis is a chronic vascular disease characterized by the deposition of lipids within the vascular wall, which triggers inflammatory responses and the formation of foam cells, ultimately leading to plaque development and vascular stenosis [42]. Increasing evidence suggests that macrophage autophagy plays a protective role in the progression of atherosclerosis,

and the pharmacologically active compounds such as NGR1 and SSB2 (derived from *Panax notoginseng* and *Bupleurum*, respectively) may enhance autophagic flux, providing a promising therapeutic strategy to mitigate the disease progression and stabilize plaques [43]. However, the synergistic mechanisms of their combined use (NS) remain unclear. In this study, we employed an integrative approach combining network pharmacology, *in vivo* ApoE$^{-/-}$ mouse model, and in vitro macrophage assays to demonstrate, for the first time, that NS combination therapy synergistically suppresses the PI3K/Akt/mTOR signaling pathway, thereby restoring macrophage autophagy flux. This dual-targeted mechanism effectively attenuates lipid accumulation, inflammation, and apoptosis, offering a novel therapeutic strategy for atherosclerosis.

Our network pharmacology analysis identified shared targets of NGR1 and SSB2 within pathways critical to atherosclerosis progression, including lipid, apoptosis, and inflammatory (Figs 1B-C). Consistent with these predictions, NS treatment in ApoE$^{-/-}$ mouse model revealed that NS combination therapy significantly improved dysregulated lipid metabolism by reducing serum TC, TG, LDL-C levels while simultaneously increasing HDL-C levels (Figs 2B-E). Histological analyses further revealed that compared to the model group and treatments with either NGR1 or SSB2 alone, NS combination therapy significantly reduced aortic plaque area and decreased the formation of hepatic fat vacuoles (Figs 3A-B and 2I), underscoring its superior efficacy in mitigating systemic lipid dysregulation. Furthermore, *in vitro* experiments showed that Oil Red O staining results indicated a marked reduction in intracellular oil droplet numbers in the NS combined treatment group relative to the model group and single-agent treatments with NGR1 or SSB2, suggesting that NS effectively inhibits lipid accumulation (Fig 4C), aligning with recent reports that macrophage lipid uptake exacerbates plaque instability [44]. Mechanistically, NS significantly enhanced the transcription levels of *PPARγ, LXRα, ABCA1*, and *ABCG1*, as determined by real-time quantitative PCR(Figs 4E-H). The reverse cholesterol transport (RCT) pathway involving PPARγ/LXRα/ABCA1 represents a potential therapeutic target for the treatment of atherosclerotic cardiovascular diseases [45], it plays a key role in RCT by promoting the transport of cholesterol from the intracellular to the extracellular space [46]. Not only that, NS inhibited NLRP3 inflammasome activation, as demonstrated by reduced mRNA and protein levels of TNF-α, IL-1β, IL-18, and NLRP3 (Fig 5). During the progression of atherosclerosis, macrophages extensively uptake oxidized low-density lipoprotein (ox-LDL) to form foam cells [47]. These foam cells are not only rich in lipids and cholesterol but also promote persistent local inflammatory responses by releasing inflammatory mediators [48]. Upon sensing danger signals such as ox-LDL, macrophages activate the NLRP3 inflammasome, which subsequently facilitates the maturation and release of pro-inflammatory cytokines IL-1β and IL-18 [49]. This anti-inflammatory effect is critical, given the established role of NLRP3 in driving atherosclerosis through IL-1β/IL-18-mediated vascular inflammation Notabley, NS combination therapy NS combination therapy restored the balance of apoptotic regulators, upregulating the expression of the anti-apoptotic protein Bcl-2 while downregulating the expression of pro-apoptotic proteins Bax and Caspase-3 (Fig 6). These findings align with emerging evidence that macrophage apoptosis promotes necrotic core formation and plaque rupture [50], positioning NS as a modulator of plaque stability. Moreover, CAP3 was also identified as a core target. Protecting macrophages from apoptosis is considered an effective strategy to mitigate plaque instability and combat acute vascular events [51]. Liao and her team proposed that macrophage apoptosis could render plaques unstable, leading to inflammation and necrosis, which increases the risk of plaque rupture and thrombosis, consequently resulting in acute cardiovascular events such as myocardial infarction or stroke [52]. Oxidative stress experiments revealed that NS combination therapy significantly reduced intracellular ROS levels. These findings indicate that the NS combination therapy markedly improves the pathological progression of atherosclerosis through synergistic effects from multiple avenues.

Central to our findings is the identification of the PI3K/AKT/mTOR pathway as a key regulatory node for NS-mediated autophagy (Figs 7−8). The mechanistic target of rapamycin (mTOR) exists in two distinct complexes: mTOR complex 1 (mTORC1) and mTOR complex 2 (mTORC2) [53], these complexes are capable of receiving signals from various pathways. The PI3K/AKT pathway is closely associated with processes such as cell growth, metabolism, inflammation, apoptosis, and autophagy [54]. As a downstream target of the PI3K/AKT pathway, the excessive activation of the PI3K/AKT/mTOR signaling cascade can promote the formation and progression of atherosclerotic plaques by inhibiting autophagic processes [55–57]. Interestingly, enrichment analysis has revealed that NGR1 and SSB2 may participate in the regulation

of atherosclerosis through their roles in lipid, apoptosis, inflammatory, and autophagy. This further corroborates the significance of the PI3K/AKT/mTOR pathway in the pathological progression of atherosclerosis. Furthermore, our *in vitro* experiments revealed that compared to the model group or monotherapy group, NS combined treatment significantly inhibited the activation of this pathway. By suppressing p-PI3K/PI3K, p-Akt/Akt, and p-mTOR/mTOR ratios (Fig 7), NS alleviated ox-LDL-induced autophagy inhibition, as evidenced by elevated LC3-II/LC3-I and Beclin-1 alongside reduced P62 (Fig 8). This mechanistic insight extends prior work demonstrating mTOR as a negative regulator of autophagy in atherosclerosis [58], and highlights NS as a potential dual inhibitor of PI3K/Akt/mTOR signaling.

In recent years, scientists have discovered that autophagy plays a positive role in atherosclerosis, a finding that has been extensively explored and emphasized in numerous studies [59–61]. In our study, the NS combination therapy significantly reversed the effects observed in the ox-LDL-induced RAW264.7 macrophage model. This was evidenced by a notable increase in the LC3-II/LC3-I ratio and Beclin-1 protein expression, along with a significant reduction in P62 protein accumulation.The high expression of the autophagy key protein Beclin-1 serves as a marker for the activation of autophagosomes [62]. Alongside LC3-II, it is considered a critical component involved in the formation of mammalian autophagosomes [63].LC3, or Microtubule-associated Protein 1A/1B-light chain 3 (MAP1LC3), is a crucial biomarker for identifying the autophagy process. When autophagy is activated, there is a significant conversion of LC3-I to LC3-II, making it an important indicator for detecting the initiation of autophagy. P62 is a connective protein that plays a crucial role in the identification and clearance of cellular cargo. Its accumulation is typically associated with a reduction in autophagic activity [64]. Conversely, during the activation of autophagy, P62 undergoes degradation, leading to a reduction in its expression levels. Our study revealed a significant increase in colocalization between p-mTOR and LAMP-1 in ox-LDL-stimulated RAW264.7 macrophages (Fig 8E). This enhanced colocalization suggests that lipid overload leads to hyperactivation of the PI3K/AKT/mTOR signaling pathway, promoting mTORC1 recruitment and anchoring to the lysosomal membrane. Previous studies have demonstrated that lysosomal localization of mTORC1 is essential for its biological functions in promoting cell growth and inhibiting autophagy [65]. As a highly glycosylated lysosomal membrane marker, LAMP-1 not only serves as a key indicator of lysosomal biogenesis [66], but also plays a crucial role as a mediator in autophagosome-lysosome fusion, which is critical for completing autophagic flux [67]. However, when mTOR becomes overactivated and accumulates on LAMP-1-positive lysosomal membranes, it phosphorylates and inhibits ULK1 (a key kinase involved in autophagy initiation), thereby blocking autophagic progression [68]. This inhibitory effect may impair the clearance of damaged lipids and protein aggregates, exacerbating intracellular lipid accumulation and ultimately promoting macrophage transformation into foam cells. This discovery reveals the significant role of NS combination therapy in regulating autophagy to prevent atherosclerosis. It is noteworthy that the expression of P62 protein in the NS combination therapy group was significantly lower than that in the monotherapy group, while the expression of Beclin-1 protein was markedly higher than that in the NGR1 monotherapy group. This indicates that the combined administration of NGR1 and SSB2 effectively modulates the expression of autophagy-related proteins through a synergistic mechanism. Therefore, this result further validates that combination therapy is more effective than monotherapy, demonstrating a more pronounced synergistic effect. In another words, the combined treatment may play a synergistic role in modulating autophagy-related proteins.

The functional significance of NS-induced autophagy was further validated using pharmacological modulators. Co-treatment with rapamycin (mTOR inhibitor) enhanced NS-mediated lipid clearance (Fig 9A, C–F), while 3-MA (autophagy inhibitor) abolished these effects, confirming autophagy as the primary mechanism (Figs 9H-O). Similarly, there was a corresponding difference in apoptosis rates(Fig 9B). Rapamycin is an mTOR inhibitor that exerts its effects by directly binding to mTORC1, thereby inhibiting its activity [69]. This action relieves the suppressive effect of mTOR on autophagy, promoting the activation of this cellular process. While 3-MA is a PI3K inhibitor that blocks the initiation of autophagy by suppressing the activity of PI3K [70]. The two compounds mentioned here are frequently utilized in atherosclerosis research. Rapamycin exerts its anti-atherosclerotic effects by inhibiting mTOR, which restores autophagic activity, reduces serum LDL cholesterol levels, alleviates inflammatory responses within the vascular system, and suppresses apoptotic processes [71]. In contrast,

3-MA inhibits PI3K to block autophagy, which exacerbates lipid accumulation and inflammatory responses, further promoting plaque instability [72]. Here, following the addition of rapamycin, there was a further reduction in the expression levels of NLRP3 and p62 compared to the NS combination treatment group. Conversely, the expressions of Bcl2 and Beclin1 were significantly elevated. In contrast, the introduction of 3-MA resulted in a marked increase in NLRP3 levels while concurrently decreasing the Beclin1 and LC3-II/LC3 ratio. As shown in Fig 9P, our immunofluorescence co-localization analysis revealed that rapamycin-treated RAW264.7 cells exhibited significantly reduced co-localization of p-mTOR with LAMP-1 compared to the normal control group. This reduction islikely attributable to rapamycin-mediated inhibition of mTOR activity which consequently impaired autophagosome-lysosome fusion. Conversely, 3-MA treatment markedly enhanced the p-mTOR/LAMP-1 co-localization, potentially through its inhibitory effect on the initial stages of autophagy, which secondarily modulates mTOR activity, thereby promoting the association between p-mTOR and LAMP-1. In summary, NS influences autophagy through the PI3K/Akt/mTOR pathway, significantly regulating the interplay between lipid accumulation, apoptosis, and inflammatory responses [73,74]. Our study elucidates a previously unrecognized synergy between NGR1 and SSB2 in combating atherosclerosis through PI3K/Akt/mTOR pathway inhibition and autophagy restoration.

## 5 Conclusion

This pioneering study provides compelling evidence that the combination of NGR1 and SSB2 (NS) effectively mitigates inflammation and inhibits the activation of the PI3K/Akt/mTOR signaling pathway in atherosclerosis. Our findings reveal that NS treatment promotes autophagy, reduces the formation of inflammatory foam cells, and decreases macrophage apoptosis (Fig 10). This multifaceted approach underscores NS's potential as a therapeutic strategy for preventing the

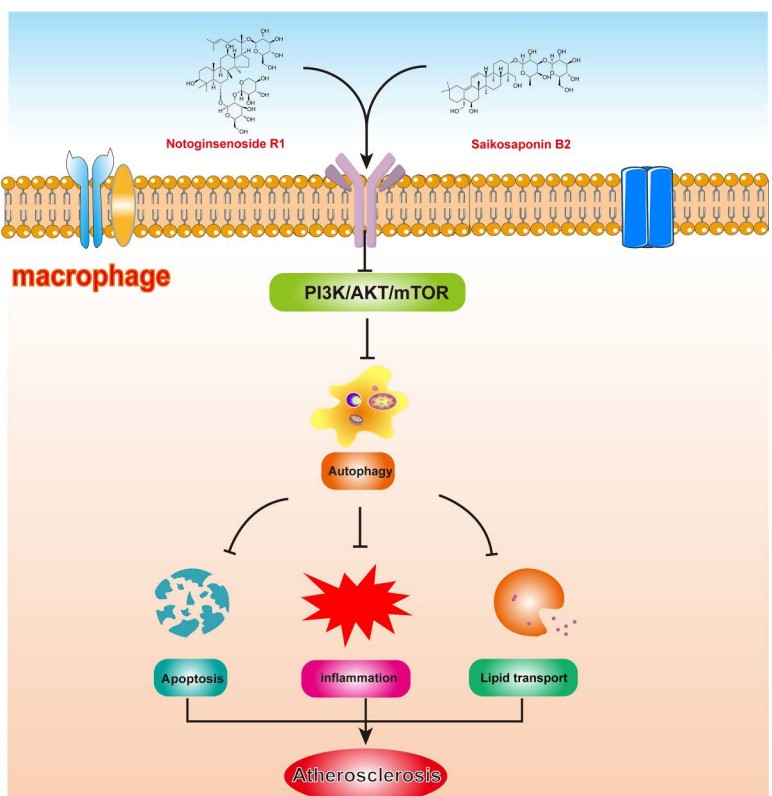

**Fig 10. NS effectively modulates the PI3K/Akt/mTOR signaling pathway and inhibits lipid transport, inflammation, and apoptosis in macrophages by enhancing autophagy.**

onset and progression of atherosclerosis. Importantly, our research elucidates the molecular mechanisms underlying the synergistic effects of NGR1 and SSB2, highlighting their therapeutic promise in atherosclerosis management. By restoring autophagic flux and modulating inflammatory responses, NS not only addresses key pathological features of atherosclerosis but also opens avenues for innovative treatment strategies targeting lipid metabolism and macrophage function.

In conclusion, this study lays a solid foundation for future investigations into the clinical applications of NS in atherosclerosis and emphasizes the need for further research to explore its potential in combination therapies. The insights gained from this research contribute significantly to our understanding of atherosclerosis pathophysiology and may inform the development of novel interventions aimed at improving cardiovascular health.

## Supporting information

**S1 Fig. Histopathological analysis of mouse kidney tissue under different treatments.** The HE-stained section of mouse kidney, scale bar = 50 μm.
(TIF)

**S1 File. S1 raw images.**
(PDF)

## Author contributions

**Conceptualization:** Chunyang Zhou.

**Data curation:** Yihua Wang, Qing Liao.

**Formal analysis:** Yihua Wang, Xue Mei.

**Funding acquisition:** Chunyang Zhou, Xue Mei.

**Investigation:** Chunyang Zhou.

**Methodology:** Yihua Wang, Qing Liao, Han Xu.

**Project administration:** Chunyang Zhou, Yihua Wang.

**Software:** Yihua Wang.

**Validation:** Yihua Wang.

**Visualization:** Yihua Wang, Rong Huang, Yue Tang.

**Writing – original draft:** Yihua Wang.

**Writing – review & editing:** Chunyang Zhou, Xue Mei, Lijun Luo.

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
