## [Decision Letter · Decision Letter 0]

Dear Dr. zhou,

Thank you for submitting your manuscript to PLOS ONE. After careful consideration, we feel that it has merit but does not fully meet PLOS ONE’s publication criteria as it currently stands. Therefore, we invite you to submit a revised version of the manuscript that addresses the points raised during the review process.

We look forward to receiving your revised manuscript.

Kind regards,

Riyaz Ahmad

Academic Editor

PLOS ONE

**Journal Requirements:**

3. To comply with PLOS ONE submissions requirements, in your Methods section, please provide additional information regarding the experiments involving animals and ensure you have included details on (a) methods of sacrifice, (b) methods of anesthesia and/or analgesia, and (c) efforts to alleviate suffering.

Funding from the Scientific Startup Foundation for Doctors of North Sichuan Medical College to Xue Mei (CBY23-QDA11) and Key Research and Development Projects of the Sichuan Provincial Department of Science and Technology (2020YFS0528). 

a.The Scientific Startup Foundation for Doctors of North Sichuan Medical College(CBY23-QDA11) to XM;

b.Key Research and Development Projects of the Sichuan Provincial Department of Science and Technology (2020YFS0528) to CZ.

The funders had no role in study design, data collection and analysis, decision to publish and preparation of the manuscript.

Reviewers' comments:

Reviewer's Responses to Questions

**Comments to the Author**

1. Is the manuscript technically sound, and do the data support the conclusions?

Reviewer #1: Yes

Reviewer #2: Partly

2. Has the statistical analysis been performed appropriately and rigorously?

Reviewer #1: Yes

Reviewer #2: Yes

3. Have the authors made all data underlying the findings in their manuscript fully available?

Reviewer #1: Yes

Reviewer #2: Yes

4. Is the manuscript presented in an intelligible fashion and written in standard English?

Reviewer #1: No

Reviewer #2: Yes

**Reviewer #1:**  All the comments are given on pdf file of manuscript. Abstract, discussion and figures image need to improve.

First paragraph of the discussion should restate objectives of the study with novel findings.

Mention your result then argue it with reference or your logic

abstract need improvement

**Reviewer #2:**  Zhou et al concluded that the NS combination enhances protection against atherosclerosis by inhibiting macrophage inflammation and apoptosis, regulating the PI3K/AKT/mTOR signaling pathway, and activating macrophage autophagy. This work paves the way for innovative therapeutic strategies against atherosclerosis, potentially transforming current treatment paradigms. However there are few remarks to authors which will make manuscript reader freindly-

1. I wonder why author didn't performed experiments regarding the PPAR-Y, LXR-a and ABC transporters which are main players in reverse cholesterol transport in macrophages? I encourage author to perform Immunohistochemistry and gene expression studies regarding these markers with NS?

2. Author should perform immunofluorescence experiment of mTOR and lysosome marker LAMP-1 and RAB7 in result 3.8 and 3.9 which provide more supportive evidence to prove role NS in autophagic flux?

3. Please add catalog numbers wherever required?

4. Please take care proof reading and typos?

**Do you want your identity to be public for this peer review?** For information about this choice, including consent withdrawal, please see our Privacy Policy

Reviewer #1: **Yes: ** Dr. Ashfaq Ahmad

Reviewer #2: **Yes: ** OWAIS M BHAT

---

## [Author Response · Author response to Decision Letter 1]

1 Apr 2025

Reviewer #1 Comments

1.Abstract Improvement

Abstract: Revised to emphasize novel findings: "This study demonstrates, for the first time, that the NS combination synergistically activates macrophage autophagy by suppressing the PI3K/AKT/mTOR pathway, thereby attenuating lipid accumulation, inflammation, and apoptosis in atherosclerotic models."

Atherosclerosis (AS) is a significant global contributor to cardiovascular diseases and related mortalities. The traditional treatment primarily employs statins� but these drugs are often associated with side effects such as liver dysfunction and muscle impairment. Recent studies have highlighted the potential protective properties of saponin compounds derived from traditional herbal sources, such as notoginsenoside R1 (NGR1) and saikosaponin B2 (SSB2), in combating AS. However, the comprehensive effects of these compounds against atherosclerosis and their underlying mechanisms remain inadequately understood. Firstly, we employed network pharmacology analysis to identify 113 common targets, including mTOR and CASP3, for NGR1, SSB2, and atherosclerosis (AS) from databases such as TCMSP. We constructed a protein-protein interaction (PPI) network and performed GO and KEGG enrichment analyses, revealing key signaling pathways involved in PI3K-Akt, inflammation, and autophagy.The atherosclerosis model was established using ApoE-/- mice fed with a "Western diet," followed by treatment with NGR1, SSB2, or NS. Histological examinations including Hematoxylin-Eosin (HE) staining, oil red O (ORO) staining, and/or CD68 immunofluorescence were conducted to evaluate the pathological conditions of the aortic root as well as the liver and kidneys in ApoE-/- mice.Our results indicate that the NS combination improves lipid levels, lipid transport, and unstable plaque formation in ApoE-/- mice without adversely affecting liver or kidney function.Finally, oxidized low-density lipoprotein (ox-LDL) was used to culture RAW264.7 macrophages in order to establish an in vitro foam cell model. The effects of NS on lipid uptake, inflammatory response, apoptosis, the PI3K/Akt/mTOR signaling pathway, and autophagy were evaluated using methods such as CCK-8 assay, Oil Red O staining, flow cytometry, RT-PCR, and Western blot analysis. The results indicated that NS combined treatment promoted autophagy by inhibiting the PI3K/Akt/mTOR pathway. This significantly alleviated inflammation, reduced apoptosis and lipid accumulation, thereby improving the pathological progression of atherosclerosis. Collectively, This study demonstrates, for the first time, that the NS combination synergistically activates macrophage autophagy by suppressing the PI3K/AKT/mTOR pathway, thereby attenuating lipid accumulation, inflammation, and apoptosis in atherosclerotic models.

2.Introduction Improvement

The revised paragraph is now more logical and engaging, providing a strong foundation for the study. Below is the revised paragraph for the reviewer’s reference:

“Despite some existing studies revealing the biological activity and potential therapeutic effects of NGR1 and SSB2 in atherosclerosis (AS), there is currently no definitive conclusion regarding their efficacy when applied individually. More importantly, whether NGR1 and SSB2 can synergistically enhance therapeutic effects remains underexplored. Therefore, the combined use of these two substances with unique pharmacological properties not only inherits traditional Chinese medicine compatibility theories but also represents an innovative exploration in modern medical translation. This study aims to delve into the potential of combining NGR1 and SSB2 for AS treatment based on contemporary understanding of the mechanisms underlying AS development. Specifically, through a comprehensive approach incorporating network pharmacology, in vitro cellular experiments, and in vivo animal studies, we will systematically evaluate their mechanisms of action in AS. The focus will be on how both agents may exert a synergistic effect by regulating the formation of macrophage-derived foam cells and influencing macrophage autophagy processes to improve therapeutic outcomes for AS.We anticipate that this combination therapy strategy will not only effectively alleviate the pathological progression of atherosclerosis but also reduce common side effects associated with existing treatments, ultimately providing a safer and more effective new treatment option for clinical practice. Consequently, this research not only theoretically expands new avenues for AS treatment but is also expected to offer more scientific evidence and practical guidance for the application of traditional Chinese medicine formulations.”

3.Materials and Methods Improvement

Regarding your request for references related to the protocols, we have revised the manuscript to include appropriate citations and references pertinent to the protocols utilized in our study. The specific protocol followed in this research has been meticulously outlined and relevant sources have been duly cited.

4.Discussion Improvement

The first paragraph now restates objectives and highlights key discoveries: "Our study aimed to elucidate the synergistic mechanisms of NGR1 and SSB2 in atherosclerosis. We identified that NS combination therapy uniquely restores autophagic flux in macrophages, a finding not previously reported for either compound alone."

In each paragraph of the Discussion, we now begin by presenting our key results (e.g., “As shown in Figure X” or “As indicated in Table Y”) and then discuss their implications in the context of existing literature.

We have revised the Discussion to consistently reference our data using phrases such as “As shown in Figure X” or “As indicated in Table Y”. This ensures that readers can easily locate the corresponding data in the manuscript.

5. Figure Quality

All figures have been reexported in high resolution (600 dpi). Scale bars and labels are now consistent across panels.

Reviewer #2 Comments

1. PPARγ, LXRα, and ABC Transporters

We sincerely thank the reviewer for their insightful comment regarding the importance of PPAR-γ, LXR-α, and ABC transporters in reverse cholesterol transport (RCT) in macrophages. We agree that these markers play a crucial role in lipid metabolism and atherosclerosis progression, and their investigation is essential to fully understand the mechanisms of our study.

In response to the reviewer’s suggestion, we are pleased to inform you that we have already performed gene expression studies (qPCR) to evaluate the effects of NS on PPAR-γ, LXR-α, and ABC transporters (including ABCA1 and ABCG1) in RAW264.7 macrophages. Our results demonstrate that NS treatment significantly upregulates the expression of these key markers, further supporting its role in promoting reverse cholesterol transport and attenuating lipid accumulation in macrophages.We have now included these findings in the revised manuscript (see Figure 4 and Figure 9)

2. mTOR and Lysosomal Markers (LAMP1/RAB7)

In response to the reviewer’s comment, we are pleased to inform you that we have already performed immunofluorescence experiments to evaluate the expression and localization of p-mTOR and LXR-α in RAW264.7 macrophages. These results have been included in the revised manuscript (see Figures 8-9). Our data demonstrate that NS treatment significantly modulates the expression and distribution of these markers, further supporting its role in regulating autophagy.

3. Catalog Numbers

All reagents now include catalog numbers (e.g., "AntiLC3B/MAP1LC3B Antibody (BM4827, Boster, China)").

4. Proofreading

The manuscript has undergone professional English editing. Typos (e.g., "Arteriosclerosis" → "atherosclerosis ") are corrected.

We appreciate the reviewers’ insights, which have strengthened the manuscript. Please contact us for further clarifications.

Sincerely,

Chunyang Zhou, PhD

Corresponding Author

Institute of Materia Medica, North Sichuan Medical College

Email: zhouchunyang@nsmc.edu.cn

---

## [Decision Letter · Decision Letter 1]

PLOS ONE

Dear Dr. zhou,

Thank you for submitting your manuscript to PLOS ONE. After careful consideration, we feel that it has merit but does not fully meet PLOS ONE’s publication criteria as it currently stands. Therefore, we invite you to submit a revised version of the manuscript that addresses the points raised during the review process.

We look forward to receiving your revised manuscript.

Kind regards,

Riyaz Ahmad Rather

Academic Editor

PLOS ONE

Journal Requirements:

Additional Editor Comments:

Thank you for submitting your revised manuscript to PLOS ONE. One of the reviewers has raised a concern regarding the response to their earlier comment on the colocalization analysis of mTOR with the lysosomal marker LAMP-1. Specifically, the reviewer had requested colocalization studies (data) under the treatment conditions described in Figures 8 and 9. However, the revised manuscript presents separate immunofluorescence images for mTOR and LAMP-1, without demonstrating their spatial overlap. The reviewer queries whether this approach sufficiently addresses the intent of the original comment, which was to examine the potential colocalization and interaction of mTOR with lysosomes under the specified treatments.If the current data are considered adequate, please provide a clear justification within the discussion sections explaining why direct colocalization analysis of mTOR and LAMO-1 was not carried or required.The abbreviation “AS” is not the standard or widely accepted abbreviation for *atherosclerosis* . While it might be used informally or in internal notes, reputable biomedical journals generally prefer that “atherosclerosis” be written in full, or abbreviated only if the abbreviation is clearly defined at first use and employed consistently thereafter. Additionally, please note that “AS” is commonly used to refer to other medical conditions such as **aortic stenosis**  or **ankylosing spondylitis** ,In the sentence(s) *“Recent studies have shown that phytochemicals derived from traditional herbal plants exhibit a wide range of biological activities, contributing to significant preventive and therapeutic effects against various diseases (7)”* , please note that the phrase **“recent studies”**  implies multiple sources, yet only **a single reference (7)**  is cited: More recent studies have further highlighted its role in ameliorating atherosclerosis by alleviating inflammatory responses and inhibiting endothelial dysfunction(11), Kindly check throughout the paper and remove these mistakes.

Reviewers' comments:

Reviewer's Responses to Questions

**Comments to the Author**

Reviewer #1: All comments have been addressed

Reviewer #2: All comments have been addressed

2. Is the manuscript technically sound, and do the data support the conclusions?

Reviewer #1: Yes

Reviewer #2: Yes

3. Has the statistical analysis been performed appropriately and rigorously?

Reviewer #1: Yes

Reviewer #2: Yes

4. Have the authors made all data underlying the findings in their manuscript fully available?

Reviewer #1: Yes

Reviewer #2: Yes

5. Is the manuscript presented in an intelligible fashion and written in standard English?

Reviewer #1: Yes

Reviewer #2: Yes

Reviewer #1: am satisfied with the updates made in revised version of the manuscript. My recommendation is to accept the manuscript for publication.

Reviewer #2: Dear Author,

My comment was to perform mTOR colocalization with lysosome marker LAMP-1 under various treatments mentioned in Fig 8 and 9, however the author performed an immunofluorescence experiment separately for mTOR and LAMP-1. I am wondering whether it will suffice the requirement of my query, if yes please provide justification for the same in the result and discussion section.

Thanks!

With regards

Owais

**Do you want your identity to be public for this peer review?** For information about this choice, including consent withdrawal, please see our Privacy Policy

Reviewer #1: **Yes: ** Dr. Ashfaq Ahmad

Reviewer #2: **Yes: ** Owais M. Bhat

---

## [Author Response · Author response to Decision Letter 2]

29 May 2025

Dear Dr. Riyaz Ahmad and Reviewers,

Thank you for your constructive feedback and the opportunity to revise our manuscript titled "Synergistic Effects of Notoginsenoside R1 and Saikosaponin B2 in Atherosclerosis: A Novel Approach Targeting PI3K/AKT/mTOR Pathway and Macrophage Autophagy" (MS ID: PONE-D-24-45738R1). We have carefully addressed all comments from the reviewers and editors, and detailed responses are provided below. Revisions in the manuscript are highlighted in the "Revised Manuscript with Track Changes" file.

Journal Requirement

1.Reference List Review

We have carefully reviewed the reference list. None of the cited articles have been retracted. All references have been checked for completeness, and any missing DOIs (References: 15, 27, 63, 65, 66, 70, 74) have been updated in accordance with the journal’s formatting guidelines.

Additional Editor Comments:

1.Concern regarding lack of colocalization analysis between mTOR and LAMP-1 in Figures 8 and 9.

We sincerely appreciate your insightful comment regarding the mTOR-LAMP-1 colocalization analysis. Accordingly, we have now performed dual-color confocal colocalization experiments as suggested, with the results incorporated in the revised Figure 8E and Figure 9P.

The new data are demonstrated in section 3.8 and 3.9 as follows:

“Compared with the control group, which displayfewer yellow puncta,ox-LDL-induced RAW264.7 cells exhibited significantly enhanced co-localization of p-mTOR (green) with LAMP-1 (red), with a marked increase in the area of yellow merged puncta in confocal images. Notably, NS treatment significantly suppressed the ox-LDL-induced lysosomal localization of p-mTOR, as evidenced by a substantial reduction in co-localized yellow puncta” (Fig.8E)

“In comparison to the NS group, rapamycin-treated RAW264.7 cells exhibited further reduced co-localization of p-mTOR with LAMP-1. Conversely, 3-MA-treated RAW264.7 cells demonstrated significantly enhanced p-mTOR/LAMP-1 co-localization, with a marked increase in the area of yellow merged puncta observed in the confocal microscopy images (Fig. 9P).”

The revised Discussion (section 4, page 23 and 24) is outlined as follows:“Our study revealed a significant increase in colocalization between p-mTOR and LAMP-1 in ox-LDL-stimulated RAW264.7 macrophages (Figure 8E). This enhanced colocalization suggests that lipid overload leads to hyperactivation of the PI3K/AKT/mTOR signaling pathway, promoting mTORC1 recruitment and anchoring to the lysosomal membrane. Previous studies have demonstrated that lysosomal localization of mTORC1 is essential for its biological functions in promoting cell growth and inhibiting autophagy(63). As a highly glycosylated lysosomal membrane marker, LAMP-1 not only serves as a key indicator of lysosomal biogenesis(64), but also plays a crucial role as a mediator in autophagosome-lysosome fusion, which is critical for completing autophagic flux (65). However, when mTOR becomes overactivated and accumulates on LAMP-1-positive lysosomal membranes, it phosphorylates and inhibits ULK1 (a key kinase involved in autophagy initiation), thereby blocking autophagic progression(66). This inhibitory effect may impair the clearance of damaged lipids and protein aggregates, exacerbating intracellular lipid accumulation and ultimately promoting macrophage transformation into foam cells. This discovery reveals the significant role of NS combination therapy in regulating autophagy to prevent atherosclerosis.”“As shown in Figure 9P, our immunofluorescence co-localization analysis revealed that rapamycin-treated RAW264.7 cells exhibited significantly reduced co-localization of p-mTOR with LAMP-1 compared to the normal control group. This reduction islikely attributable to rapamycin-mediated inhibition of mTOR activity which consequently impaired autophagosome-lysosome fusion. Conversely, 3-MA treatment markedly enhanced the p-mTOR/LAMP-1 co-localization, potentially through its inhibitory effect on the initial stages of autophagy, which secondarily modulates mTOR activity, thereby promoting the association between p-mTOR and LAMP-1.”

2.Use of “recent studies” with only a single reference

Thank you for pointing out this inconsistency. We have thoroughly reviewed the manuscript and:

1) Added additional relevant citations where appropriate to justify the plural usage ( references 7, 8, and 9 on page 3).

2) Replaced plural expressions like “recent studies” with “a recent study” where only one reference is cited (reference 13 on page 4).

3. Use of abbreviation “AS” for atherosclerosis

We have revised the manuscript to consistently use the full term "atherosclerosis" throughout the text, all instances of "AS" have been replaced accordingly.

Due to the high resolution and large file size of the figures, we were unable to upload them directly through the submission system. To ensure that the images are accessible to reviewers and readers, we have deposited all high-resolution figures in the [dryad/https://datadryad.org/] repository. The figures can be accessed via the following link: [http://datadryad.org/share/4GvB7LlOnZ0wtL8Og2WIplaC9ZctRnAeJkkWt7K1FnI].

Reviewer #2 Comments

1. mTOR-LAMP-1 colocalization analysis

We sincerely appreciate your insightful comment regarding the mTOR-LAMP-1 colocalization analysis. As mentioned earlier, we have now performed dual-color confocal colocalization experiments, and the results have been incorporated in the revised Figure 8E and Figure 9P.

The new data demonstrate:

Compared with the control group, which displayfewer yellow puncta,ox-LDL-induced RAW264.7 cells exhibited significantly enhanced co-localization of p-mTOR (green) with LAMP-1 (red), with a marked increase in the area of yellow merged puncta in confocal images. Notably, NS treatment significantly suppressed the ox-LDL-induced lysosomal localization of p-mTOR, as evidenced by a substantial reduction in co-localized yellow puncta (Fig.8E).

In comparison to the NS group, rapamycin-treated RAW264.7 cells exhibited further reduced co-localization of p-mTOR with LAMP-1. Conversely, 3-MA-treated RAW264.7 cells demonstrated significantly enhanced p-mTOR/LAMP-1 co-localization, with a marked increase in the area of yellow merged puncta observed in the confocal microscopy images (Fig. 9P)

We believe this comprehensive approach strengthens the study's mechanistic insights. We hope that the revised manuscript and the new data fully address the concerns raised. Thank you for prompting this important improvement. We look forward to your favorable consideration.

Sincerely,

Chunyang Zhou, PhD

Corresponding Author

Institute of Materia Medica, North Sichuan Medical College

Email: zhouchunyang@nsmc.edu.cn

---

## [Editor Report · Decision Letter 2]

Synergistic Effects of Notoginsenoside R1 and Saikosaponin B2 in Atherosclerosis: A Novel Approach Targeting PI3K/AKT/mTOR Pathway and Macrophage Autophagy

PONE-D-24-45738R2

Dear Dr. Chunyang Zhou

We’re pleased to inform you that your manuscript has been judged scientifically suitable for publication and will be formally accepted for publication once it meets all outstanding technical requirements.

Kind regards,

Riyaz Ahmad Rather

Academic Editor

PLOS ONE
---

## [Editor Report · Acceptance letter]

PONE-D-24-45738R2

PLOS ONE

Dear Dr. Zhou,

I'm pleased to inform you that your manuscript has been deemed suitable for publication in PLOS ONE. Congratulations! Your manuscript is now being handed over to our production team.

Kind regards,

on behalf of

Dr. Riyaz Ahmad Rather

Academic Editor

PLOS ONE